Stratospheric circulation response to large Northern high-latitude volcanic eruptions in a global
climate model
Hera Guðlaugsdóttir[1,2], Yannick Peings[2], Davide Zanchettin[3], Gudrun Magnusdottir[2]
[1]University of Iceland, institute of Earth Sciences, 102 Reykjavík, Iceland
[2]University of California Irvine, Department of Earth System Science, Irvine CA 92697-3100,
United States
[3]University Ca'Foscari of Venice, department of Environmental Sciences, Informatics and
Statistics, 30123 Venice, Italy
*Correspondence to:* Hera Guðlaugsdóttir (hera@hi.is)
Abstract
Stratospheric aerosols after major explosive volcanic eruptions can trigger climate anomalies for up to
several years following such events. Whereas the mechanisms responsible for the prolonged response to
volcanic surface cooling have been extensively investigated for tropical eruptions, less is known about the
dynamical response to high-latitude eruptions. Here we use global climate model simulations of an idealized
6 month long northern hemisphere high-latitude eruption to investigate the stratospheric circulation
response during the first three post-eruption winters. Two model configurations are used, coupled with an
interactive ocean and with prescribed sea surface temperature. Our results reveal significant differences in
the response of the polar stratosphere with an interactive ocean: the surface cooling is enhanced and zonal
flow anomalies are stronger in the troposphere, which impacts atmospheric waveguides and upward
propagation of large-scale planetary waves. We identify two competing mechanisms contributing to the
post-eruption evolution of the polar vortex: 1) A local stratospheric top-down mechanism whereby
increased absorption of aerosol-induced thermal radiation yields a polar vortex strengthening via thermal
wind response; 2) A bottom-up mechanism whereby anomalous surface cooling yields a wave-activity flux
increase that propagates into the winter stratosphere. We detect unusually high frequency of Sudden
Stratospheric Warmings in the idealized forcing simulation with interactive ocean that calls for further
exploration. In the coupled runs, the top-down mechanism dominates over the bottom-up mechanism in
winter 1, while the bottom-up mechanism dominates in the follow-up winters.

## 1 Introduction

The enhancement of the stratospheric aerosol layer following strong sulfur-rich explosive volcanic eruptions is an important driver of natural climate variability due to the short-lived yet possibly very strong radiative anomalies imposed within the atmospheric column (Robock, 2000; Timmreck, 2012; Zanchettin, 2017). This direct radiative effect can alter both meridional surface and stratospheric temperature gradients that can, in turn, initiate further dynamical climate responses on seasonal to decadal time scales (Church et al., 2005; Gleckler et al., 2006; Stenchikov et al., 2009; Shindell et al., 2009; Otterå et al., 2010; Zanchettin et al., 2012; Swingedouw et al., 2015). Direct radiative and dynamical responses critically depend on the spatiotemporal characteristics of the enhanced stratospheric aerosol layer, which ultimately depends on the characteristics of the eruption, such as magnitude, timing and location (Stenchikov et al., 2009; Shindell et al., 2009; Zanchettin et al., 2012; Swingedouw et al., 2015). Spatiotemporal characteristics of volcanic aerosol from high-latitude (HL), Northern Hemisphere (NH) eruptions are typically very different when compared to tropical eruptions. Accordingly, several studies have shown that HL eruptions typically initiate different climate responses compared to tropical eruptions (Meronen et al., 2012; Pausata et al., 2015; Guðlaugsdóttir et al., 2018; Zambri et al., 2019; Sjolte et al., 2021). Therefore, tropical eruptions cannot be considered close analogs of HL eruptions, underlining the need for more studies on the latter to further quantify their potential climate impacts (Zanchettin et al., 2016). In this study we explore how stratospheric sulfate aerosol enhancements largely constrained in the NH extratropics affect hemispheric-scale atmospheric dynamics, with a focus on the stratospheric polar vortex and on temporal evolution of responses through three perturbed winters.

The winter stratospheric polar vortex is considered to play a deciding role in distinguishing between the response to low- and HL NH enhancements of the stratospheric sulfate aerosol layer, where opposite responses are expected to emerge under the same mechanism: When the stratospheric sulfate aerosol layer is enhanced at low latitudes, e.g., following tropical volcanic eruptions, local warming by infrared absorption increases the meridional stratospheric temperature gradient that can lead to a stratospheric polar vortex strengthening due to a thermal wind response (e.g., Zanchettin et al., 2012; Bittner et al., 2016). Conversely, the local warming from aerosols constrained at higher latitudes decreases the meridional temperature gradient, promoting a

weakening of the polar vortex (Kodera, 1994; Perlwitz & Graf, 1995; Stenchikov et al., 2002; Oman et al., 2005; Sjolte et al., 2021). The downward propagation of the stratospheric polar vortex anomaly into the troposphere can lead to regime shifts of the tropospheric Arctic Oscillation and associated anomalous regional surface patterns (e.g., Zanchettin et al., 2012; Zambri et al., 2017). In the case of polar vortex weakening, a critical role is attributed to increased likelihood of Sudden Stratospheric Warming events (SSWs) (Haynes, 2005; Domeisen et al., 2020; Huang et al., 2021; Kolstad et al., 2022, and references therein), whose projection on a negative Arctic Oscillation is expected to bring a series of consequences, including increased frequency of tropospheric blockings and mid-latitude cold air outbreaks (e.g., Ma et al., 2024). However, the negative Arctic Oscillation and associated tropospheric anomalies following SSWs are characterized by a low signal-to-noise ratio (e.g., Charlton-Perez et al. 2018; Zhang et al. 2019). Accordingly, recent studies tend to disagree on this top-down mechanism being a robust dominant feature of climate response to volcanic eruptions (Weierbach et al., 2023; DallaSanta and Polvani, 2023; Kolstad et al., 2022; Azoulay et al., 2021; Polvani et al., 2019; Zanchettin et al., 2022; Toohey et al., 2014). The radiative surface cooling following large volcanic eruptions has been shown to affect the stratospheric polar vortex via a bottom-up mechanism (e.g., Graf et al., 2014; Peings and Magnusdottir, 2015; Omrani et al., 2022). An example of this bottom-up mechanism following HL eruptions is demonstrated in Sjolte et al. (2021) where they linked a weak polar vortex to an increase in wave energy flux from the troposphere into the stratosphere without the meridional stratospheric temperature gradient playing a major role.

With this in mind, the importance of transient atmospheric eddies (waves) and eddy-mean flow interactions is becoming increasingly clear in explaining vertical and horizontal propagation of atmospheric perturbations of various origins (e.g. Smith et al., 2022; Nakamura, 2023). DallaSanta et al. (2019) used a hierarchy of simplified atmospheric models to show that eddy feedbacks are crucial in explaining stratosphere-troposphere coupling as well as the stratospheric response alone following a tropical Pinatubo-like eruption. This demonstrates that the anomalous atmospheric circulation response to an enhanced stratospheric sulfate aerosol layer cannot be understood as the mere adjustment to meridional temperature gradients, and that eddy-mean flow interactions and eddy feedback are an essential contribution to such response. Both mechanisms, i.e., the top-down mechanism triggered by local stratospheric heating and the bottom-up mechanism triggered by surface cooling, act together in the real world and in realistic simulations. Therefore, idealized

model experiments are required to assess their relative contribution to uncertainty in regional
climate variability during the period following the enhancement of the sulfate aerosol layer
(Zanchettin et al., 2016).
Icelandic volcanism has played a role in shaping past NH climate variability and will continue
doing so. Two Icelandic eruptions during the past 2000 years, namely Eldgjá in ~939 CE and Laki
in 1783 CE, are considered to have had a significant impact on climate variability up to the global
scale (Brugnatelli and Tibaldi, 2020; Zambri et al., 2019; Oppenheimer et al., 2018; Thordarson
and Self, 2003; Stothers, 1998). These types of effusive eruptions are common in Iceland where
their duration can extend over years. During part of the eruption time such eruptions can become
explosive (referred to as mixed-phase eruptions) when ascending magma in a conduit comes in
contact with water as was considered the case with both Eldgjá and Laki, explaining their
widespread impacts. Eruption history as well as dense monitoring network of Icelandic volcanic
systems tell us that many of these systems are currently on the verge of an eruption, having already
produced some of the largest volcanic eruptions over the past millennia (e.g., Öræfajökull,
Bárðabunga and Hekla, Larsen & Guðmundsson, 2014; Barsotti et al., 2018; Einarsson, 2019).
Therefore history and current activity makes these types of eruptions an ideal reference case to
explore the potential climatic impacts of HL enhancements of the stratospheric sulfate aerosol
layer and to test hypotheses about the underlying mechanisms driving the climate response. This
is the focal point of this study where we investigate for the first time the role of wave-mean flow
interactions and SSWs in the climate response to a HL volcanic eruption. For this we perform
idealized, long-lasting HL volcanic perturbation experiments using the Community Earth System
Model version 1 (CESM1) in its coupled and atmosphere-standalone configurations. We evaluate
the NH response during the first three winters following the eruption, referred to as post-eruption
winters in the text, and assess the dominating mechanism in each winter. This paper is organized
as follows: Section 2 describes the model, experimental design and diagnostics; results are
presented in section 3 followed by discussions in section 4 where we end with concluding our
results in section 5.

**2 Methods**
**2.1. Numerical Model**
We use the Community Earth System Model (CESM) version 1, developed by the National Center
for Atmospheric Research (NCAR). In our configuration of CESM1, the atmospheric component
is the Whole Atmosphere Community Climate Model, version 4 (WACCM4, Marsh et al. 2013).
WACCM4 includes 66 vertical levels (up to $5.1\times10^{-6}$ hPa, $\sim$140 km) and uses CAM4 physics.
We use the specified chemistry version of WACCM4 (SC-WACCM4), which is computationally
less expensive to run, but simulates dynamical stratosphere-troposphere coupling and stratospheric
variability like SSWs and the polar vortex with skills comparable to the interactive chemistry
model version (Smith et al., 2014). CESM1/WACCM4 uses the Community Atmospheric Model
Radiative Transfer (CAMRT) to parameterize the radiative forcing where it has been shown to
accurately represent stratospheric aerosols by f. ex. simulating the temperature response following
Mt. Pinatubo in 1991 (Neely et al., 2016). The SC-WACCM4 experiments are run with a horizontal
resolution of 1.9° latitude by 2.5° longitude and include present-day (year 2000) radiative forcing.
A repeating 28-month full cycle of the Quasi-biennial Oscillation (QBO) is included in the SC-
WACCM4 experiments through nudging of the equatorial stratospheric winds to observed
radiosonde data. In the coupled ocean-atmosphere configuration, the ocean component of CESM1
is the Parallel Ocean Program version 2 (POP2). CESM1 also includes the Los Alamos sea-ice
model (CICE), the Community Land Model version 4 (CLM4) and the River Transport Model
(RTM). CLM is run at a horizontal resolution of 1.9°x2.5°, POP2 and CICE are run at nominal 1°
resolution with higher resolution near the equator than at the poles. Further details about CESM1
are given in Hurrell et al. (2013).

**2.2. Volcanic Forcing File**
We use the Easy Volcanic Aerosol (EVA) forcing generator (Toohey et al., 2016). EVA provides
zonally symmetric stratospheric aerosol optical properties as a function of time, latitude, height,
and wavelength (see detailed information on the tool in Toohey et al., 2016). EVA has been used
to generate volcanic forcing in both idealized volcanic experiments (e.g., Zanchettin et al., 2016)
and realistic paleoclimate simulations (Jungclaus et al., 2017) contributing to the sixth phase of
the coupled model intercomparison project.
We use EVA to prescribe the volcanic aerosol loading corresponding to that of the 1991 Mt.
Pinatubo eruption (14.04 Tg $SO_2$), but at 45° N. Since the model reads the volcanic forcing as
aerosol mass mixing ratio (kg/kg) while our EVA output is in $1/m^2$ (aerosol extinction), we scale
our forcing file by using the standard aerosol input file for CAM4 and 5 (see Neely et al., 2016,
Table 1) for the same eruption. A monthly scaling factor was derived from this linear relationship
between the aerosol extinction and the aerosol mass mixing ratio that was used to scale the raw
EVA forcing data (Fig. 1). From these scaled forcing data, the aerosol optical properties for our
experiments are obtained with a two-step approach. First, we move the injection location
northwards so that the center of the aerosol mass is at 65° N latitude and spans 10-28 km in altitude.
Then, we define the start of the eruption to be May 1st and prolong the peak of the forcing by
extending in time the highest monthly value in the so-obtained forcing data, so that the decline in
aerosol mass begins 6 months after the start of the eruption or on November 1 (see Fig. 1). We thus
obtain aerosol optical properties for an idealized, long-lasting high-latitude NH eruption. In this
experiment we assume stratospheric injection only, although similar eruptions in the natural world
would likely inject part of the total aerosol mass within the troposphere during the eruption. Past
NH eruptions like Eldgjá and Laki had an atmospheric $SO_2$ loading of 219Tg and 122Tg
respectively that was carried aloft with the eruptive column up into the upper troposphere with
portions of the aerosols reaching the lower stratosphere during the eruptions (Thordarson et al.,
2001). Hence our experiment can also be considered as a 6-month stratospheric aerosol injection
that is analogous to similar although smaller eruptions (as compared to Laki) without the
tropospheric aerosols.

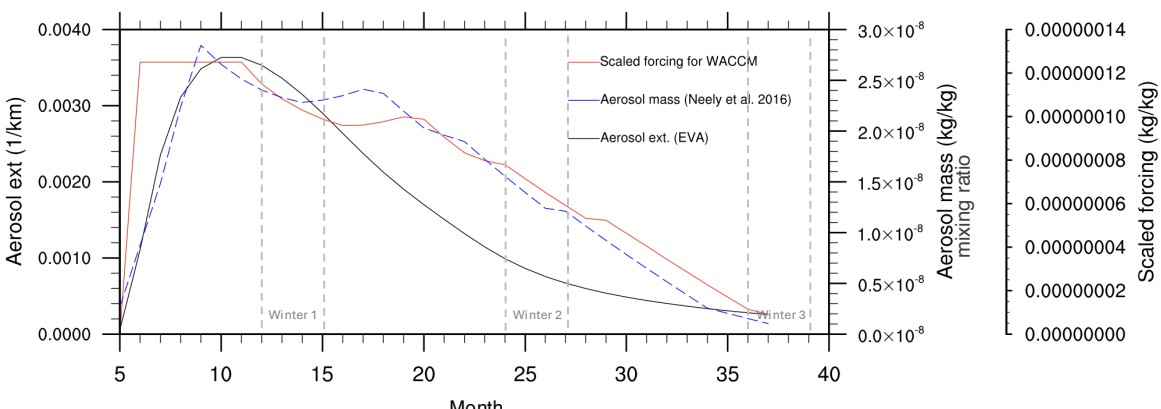


Figure 1: The time series of the original EVA aerosol extinction output (1/km, black curve) and the
dry aerosol mass mixing ratio of the volcanic forcing file of Neely et al. (2016) (kg/kg, blue dashed
curve) used for deriving the linear scaling coefficient for the conversion of EVA output into
WACCM4 input (kg/kg, red curve). The horizontal axis is time (months) from the start of the
eruption. Here we assume that the aerosol lifetime at 65° N is the same as at 45° N. Dashed vertical
lines show the three winters that we focus on in this study.

**2.3. Experimental design**
We ran two volcanic perturbation experiments with CESM1. The first experiment is conducted
with the atmosphere-only version of the model, where boundaries to SC-WACCM4 are provided
by prescribed fields of sea-surface temperature (SST) and sea-ice concentration (SIC)
corresponding to the 1979-2008 monthly climatology of HadISST observations (Rayner et al.,
2003). We refer to this experiment as *atm-only*. The second experiment is conducted with the
coupled version of the model, henceforth referred to as *cpl*. For each experiment we run 20
ensemble members including the volcanic forcing and 20 paired ensemble members without the
volcanic forcing and otherwise identical to the volcanic simulations, which we refer to as the
control.
The atmosphere-only experiments were run over three full years, which provides two full winters
after the onset of the eruption. We found that there was no need to extend the simulations further
given the duration of the forcing and short memory of the atmosphere. The coupled experiments
follow a similar protocol but they were integrated over 15 years to assess the response influenced
by oceanic dynamical adjustment. However, in this study we only focus on the first three winters
following the eruption, where the January and February forcing of winter 3 are defined to be a
continuation of the December value of year 2. We define the first post-volcanic winter as December
of the starting year (year 0) and the following January and February (year 1), the second post-
volcanic winter is then December of year 1 and the following January and February of year 2 etc.
Because the QBO is prescribed, and given its importance for the atmospheric circulation and the
distribution of volcanic aerosols within the stratosphere (Thomas et al., 2009; DallaSanta et al.,
2021; Brown et al., 2023), we have been careful in homogeneously sampling the QBO phasing
that is imposed on the 20 ensemble members. For this, we shift the 28-month QBO cycle by one
month for every ensemble member, so that the phasing of the QBO differs from one ensemble
member to the next (Elsbury et al., 2021). This avoids potential biases in the climatic response that
may be induced by any dominating QBO phase.


## 2.4. Diagnostics

Model output is analyzed by computing paired anomalies, defined as deviations of each volcanic simulation from the corresponding control simulation (Zanchettin et al., 2022) (volcanic minus control). The statistical significance of the ensemble mean of paired anomalies is assessed at the 95% confidence interval, calculated from all 20 ensemble members, using a two-sided Student's t-test in addition to a Kolmogorov-Smirnov test.

To evaluate the effects of planetary waves on the zonally-averaged stratospheric response, we use the Eliassen Palm (EP) flux and its divergence (Edmon et al., 1980) in addition to the 3D generalization of the EP flux, the Plumb flux (Plumb, 1985), for a longitudinal representation in the lower troposphere and stratosphere. We identify SSW events by using an algorithm following the procedures described in Charlton and Polvani (2007), where mid-winter sudden warming events are determined to take place if the 10 hPa zonal-mean zonal wind at 60°N becomes easterly. Once a warming is identified, no day within 20 days of a central date, defined as the first day in which the daily mean zonal-mean wind at 60N and 10hPa is easterly, can be defined as an SSW. Changes in conditions for large-scale planetary waves propagation (waveguides) are examined using the optimal propagation diagnostic for stationary planetary waves, described in Karami et al. (2016). This metric is based on the construction of Probability Density Functions for positive values of the refractive index (Matsuno 1970), as a function of zonal and meridional wave numbers. The refractive index is calculated using daily zonal wind and temperature at all levels, to derive monthly and zonally-averaged probabilities for stationary Rossby waves to propagate through the atmosphere, in function of latitude and pressure level. This is calculated for zonal wave numbers k=1,2,3 and meridional wave numbers l=1,2,3 (large-scale waves), and we average the probabilistic refractive index for each of the nine combinations of k and l, to provide a general estimate of chances for propagation of stationary planetary waves. For the eddy feedback calculations we compute the square of the local correlation across the ensemble members between DJF zonal-mean zonal wind and the divergence of the northward EP flux (delta phi F phi) averaged over 600-200 hPa (Smith et al. 2022). In addition, we compute the rate of temperature (K) changes in the 2m temperature (T2m) gradient using spherical harmonics to yield a T2m gradient in the meridional (dZ/dlat) and zonal (dZ/dlon) directions.

**3 Results**

In the following sections we will investigate the *cpl* experiment to characterize the forced response and identify the mechanism by utilizing the information provided by the *atm-only* experiment. We begin our investigation in the upper atmosphere before making our way towards the surface.

**3.1. Volcanic radiative forcing**

The net shortwave (SW) and longwave (LW) downward flux at the top of the atmosphere show an expected behavior following a stratospheric sulfate injection where we see a decrease in the SW due to scattering and an increase in the LW due to absorption around the injection location (Fig. 2c-f). Temporal perturbations of SW fluxes for both *cpl* and *atm-only* are influenced by the obvious strong seasonal evolution in solar insolation, where we see strong anomalies during the first summer north of 30° N than then becomes more confined to the mid latitudes as winter progresses with a slow decrease towards the end of the third year (Fig. 2c-d).

LW anomalies also show seasonal evolution with stronger LW flux at mid latitudes compared to at high latitudes during summer that continues into the winter season and remains significant throughout most of these three years. During winter, the LW anomalies are present at high latitudes where the SW anomalies are absent. The latitudinal bands of radiative flux anomalies correspond to the maximum values of the aerosol mass between 60 and 70° N, and the total aerosol mass of 14.04 Tg being largely confined north of 45° N (Fig. 2a-b). Overall, the idealized radiative forcing is largely bounded by the NH extratropics with the exception of a slight significant increase around 30-60° S in the second and third summer (Fig. 2c-d) that is visible at around 14-15 km a.s. (Fig. 2b). This occurs due to spatial features in the Neely et al. (2016) aerosol forcing that we use for scaling, where a slight aerosol increase occurs at lower latitudes, although this is not detectable when the aerosol mass is averaged through the atmospheric column with respect to time (Fig. 2a). We also detect a slight difference in the LW and SW fluxes that arises from differences in high cloud cover between *atm-only* and *cpl*, where *cpl* shows a decrease in high cloud cover in the northern high-latitudes, compared to *atm-only* (not shown).

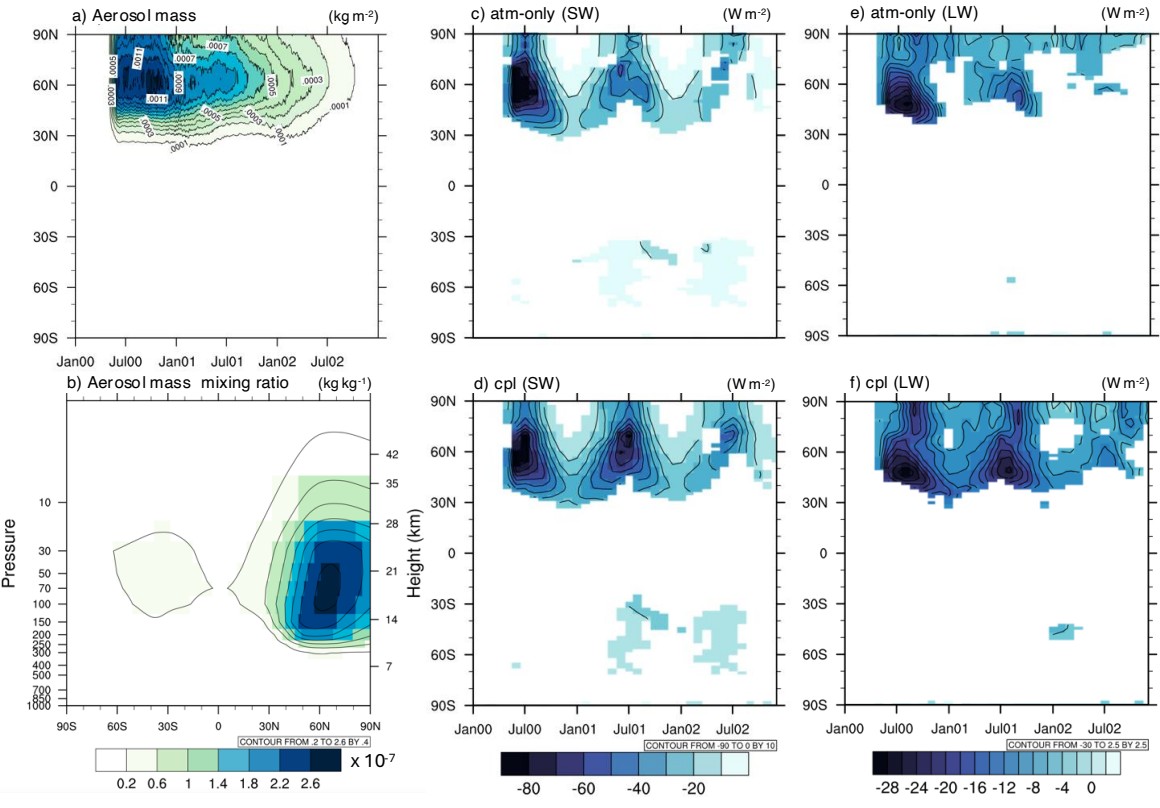

Figure 2: a) Average aerosol column mass time evolution in kg/m2 and b) pressure vs. latitude slice of the prescribed aerosol dry mass mixing ratio in kg/kg (3-year average). Aerosol mass is the same in cpl and atm-only. c) and d) show the time evolution of the net SW flux (downward) anomaly at the top of the atmosphere, and e) and f) the same but for net LW flux anomaly, resulting from the volcanic aerosol mass, in c) *atm-only* and d) *cpl* where coloured areas indicate 99% significance compared to the control experiment according to a Student's t-test.

### 3.2. Stratospheric response

The strong seasonality in the LW perturbations described above also characterizes stratospheric temperatures, where a strong increase in the zonally averaged temperature at 50 hPa (T50) is detected north of 30° N during post-eruption summers in both experiments (Fig. 3a and 3b). This summer warming is followed by a net cooling of the polar stratosphere in the first winter seen for both *cpl* and *atm-only*. A clear difference in the T50 response in the two experiments is seen in winter 2, where *cpl* yields warming over polar latitudes while *atm-only* yields cooling (Fig. 3a,b). This reveals the intra-seasonal dynamical effects in the *cpl* experiment beyond the direct radiative

response as we will see later on. The contrasting temperature response is accompanied by an
opposite response in the zonal-mean zonal winds at 10hPa (U10) between 70 and 80° N. This U10
response is an indicator of the state of the polar vortex, where a polar vortex weakening is detected
in winter 2 for *cpl* but a strengthening *atm-only* (Fig. 3c-d). Figures 3c-d do show a large ensemble
spread in the zonal mean U10 winter response that is evident of a low signal to noise ratio. While
the first winter in *cpl* and the first two in *atm-only* show little statistical significance according to
a Kolmogorov-Smirnov test, this significance does increase for winter 2 in the *cpl* experiment. We
also see this weakening in the zonal mean U50, also showing stronger significance during winter,
(Supplementary Fig. S3) but not as clearly as in the zonal mean U10. However, for consistency we
will mainly be focusing on the U50 response in the following section where this response is clear
over the NH polar cap. The difference between *cpl* and *atm-only* will be in the focus in the
following sections.

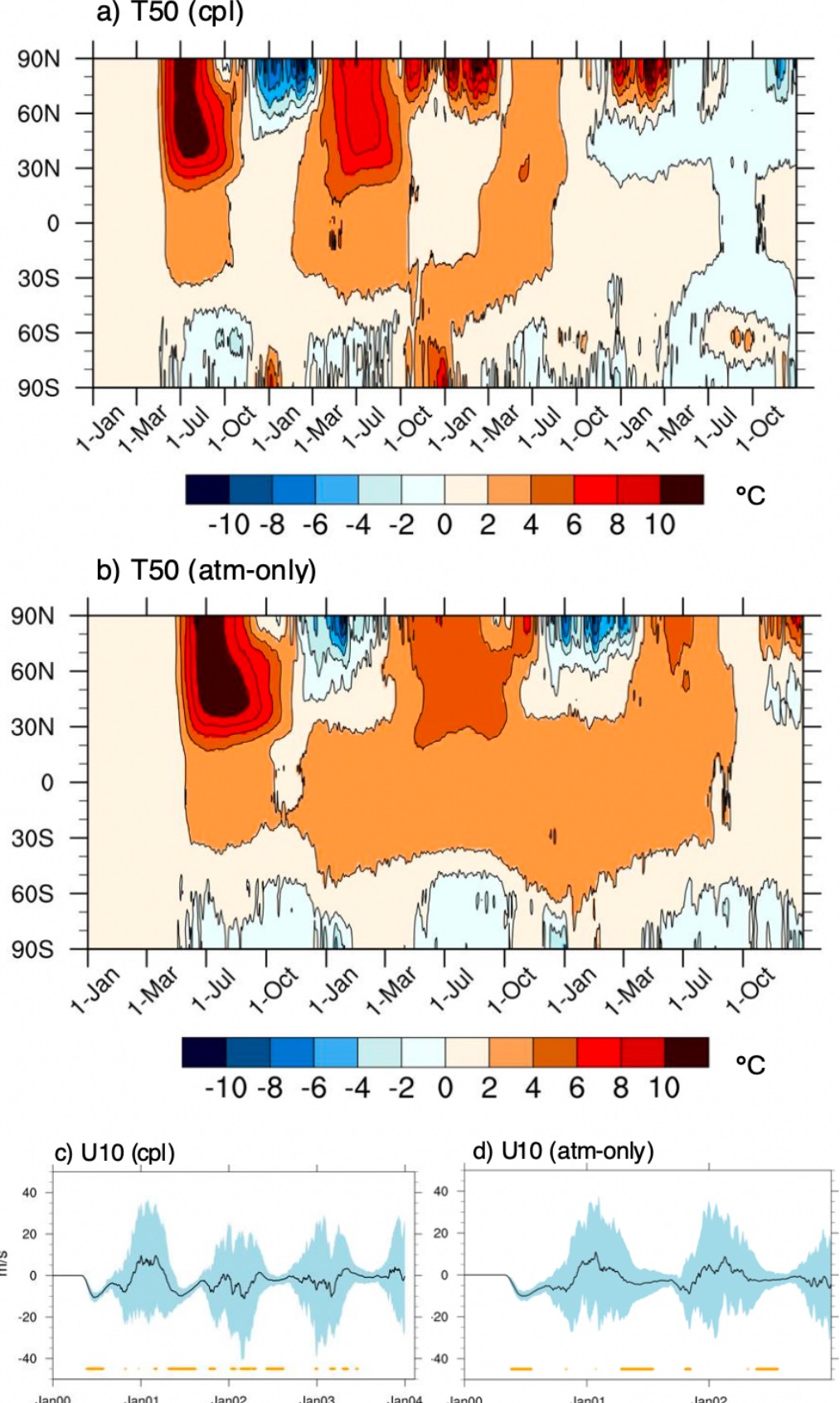


Figure 3. a,b) Latitude versus time response of T50 anomalies over the investigated period in a) *cpl* and b) *atm-only* (note the different time scale). Contours are significant in 95% confidence intervals according to a Student's t-test. c,d) Stratospheric polar vortex response shown as the zonal mean U10 anomalies between 70 and 80° N for c) *cpl* and d) *atm-only*. Black lines show the ensemble mean anomalies and blue shadings show the ensemble +/- 2 standard deviation anomaly range. Orange markers indicate when the difference between perturbed and unperturbed experiments becomes significant (p<0.05) according to a Kolmogorov-Smirnov test.

### 3.2.1 - First post-eruption winter

In the *cpl* experiment, the polar vortex strengthening in winter 1 is associated with extensive anomalies in temperature and zonal wind at 50 hPa (Fig. 4a). The anomalous temperature pattern consists of cooling at high latitudes and into the midlatitudes over the Atlantic, and warming over large swaths of the subtropics (to 20° N) and into the midlatitudes over the Pacific. This temperature pattern is also identified in the zonal mean T50 (Fig. 3b). Similarly, the zonal wind weakens into the midlatitudes over the Pacific while it is stronger in mid to high latitudes over the Atlantic. The strong upward EP flux (black arrows) is an indicator of the direction of propagated waves originating at the surface around the midlatitudes, where the horizontal and vertical EP flux components are proportional to the eddy momentum and heat flux, respectively (Fig. 4d). A convergence (negative divergence, dashed red contours) in the EP flux is detected in the upper troposphere that acts to weaken the tropospheric westerlies (Fig. 4d and Fig. S2). However, the EP flux and its convergence within the stratosphere does not appear to impact the stratospheric mean flow and the polar vortex. Therefore the local heating due to the volcanic aerosols and the associated increase in the meridional temperature gradient in the stratosphere appear to dominate the response of the polar vortex via thermal wind response, also depicted by the LW anomalies (Fig. 2f). Winter 1 in *atm-only* shows a similar thermal wind mechanism at play in the stratosphere as for the *cpl* experiment (Fig. 5a and 4a, respectively). In that case, less obvious tropospheric influences are detected, due to lack of forced surface cooling, as seen in the limited anomalous upward wave activity detected by the EP flux diagnostics (Fig. 5c).

333

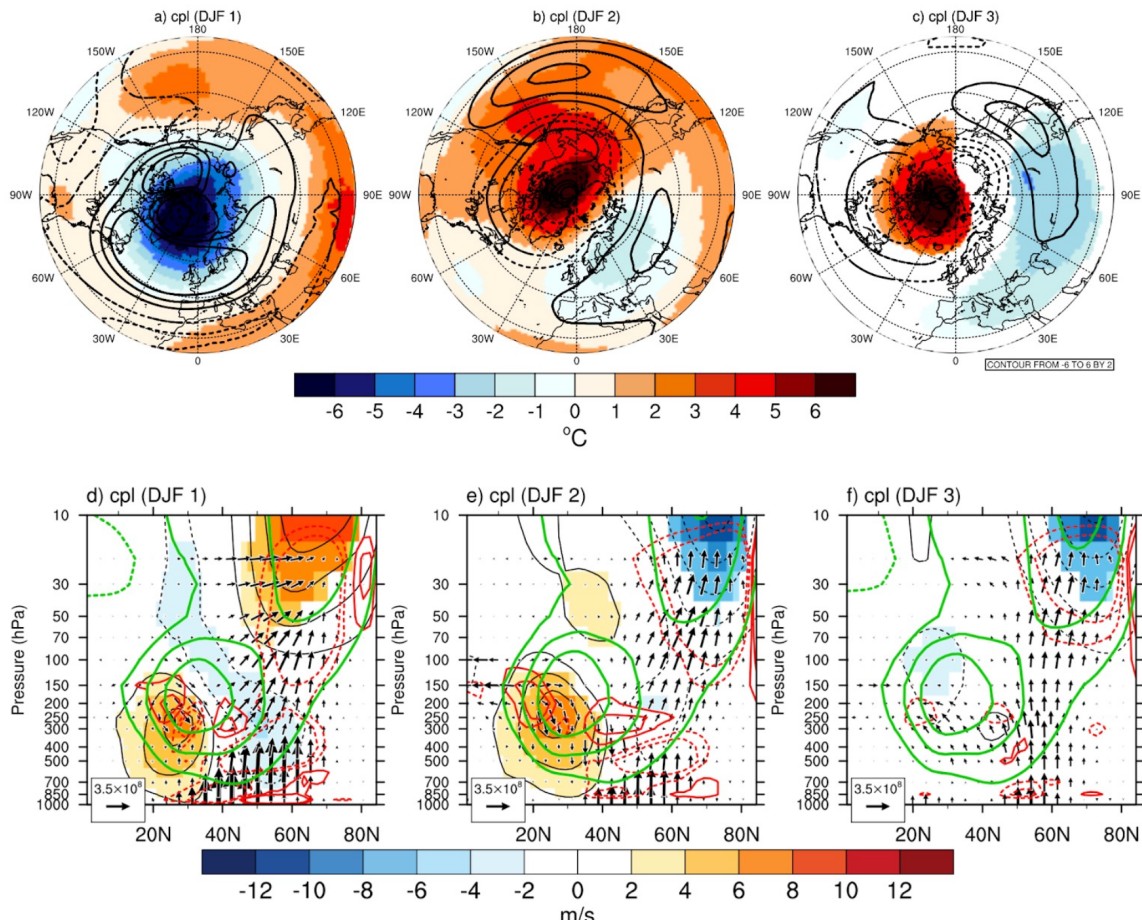

334

Figure 4: Winter stratospheric response in the *cpl* experiment. a-c) U50 (contours) and T50 (shading: red = warming, blue = cooling) response for winters 1-3, respectively. d-f) EP flux (arrows) and divergence (red contours) response, along with zonal-mean zonal wind response (black contours and shading: red = strengthening, blue = weakening) and climatology (green contours) in winters 1-3, respectively. Contours and color-shaded areas indicate 95% significance according to a Student's t-test. Only vectors that are significant at the 95% confidence interval are shown.

342

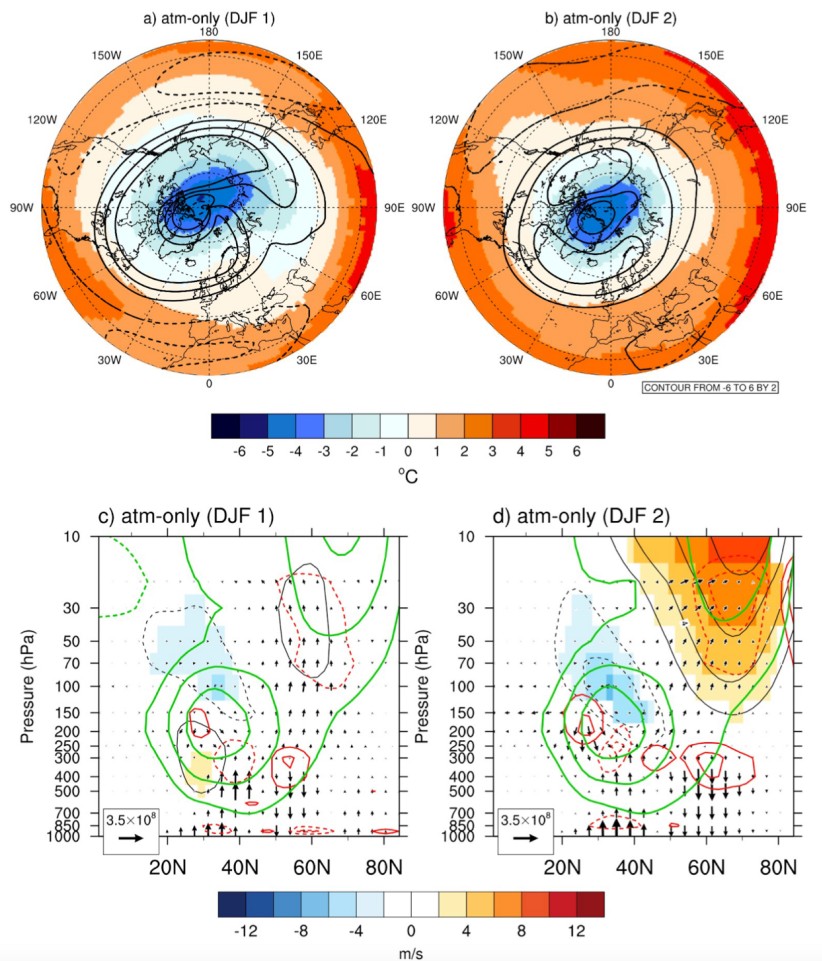

Figure 5: The same as Fig. 4 but for the *atm-only* experiment. a-b) Zonal wind (contours) and temperature (shading) response at 50 hPa for winter 1 and winter 2, respectively. c-d) EP flux (arrows) and divergence (red contours) response along with zonal-mean zonal wind (black contours) and pure climatology (green contours, 2m) in winters 1-2, respectively. Contours and colored area indicate 95% significance according to a Student's t-test.

### 3.2.2. Second post-eruption winter

A stark difference in the polar vortex response is detected between *cpl* and *atm-only* in winter 2. While *atm-only* exhibits a response similar to winter 1 (Fig. 5b), a significant warming over North America and the North Pacific emerges in *cpl* along with a weakening of U50 at high latitudes (Fig. 4b) indicating a shift of the polar vortex towards Eurasia. This warming at high latitudes then coincides with a slight LW absorption at high latitudes (Fig. 2f). The U50 weakening is not uniform throughout the longitudes explaining the lack of response detected in the zonal mean U50 (Fig.

S3), where one needs to go to U10 to get a clear response in the zonal-mean zonal wind (Fig. 3c).
An anomalously strong upward propagation of planetary waves persists in the *cpl* (Fig. 4e), with
a stronger upward EP flux now protruding into the stratosphere above 20hPa in contrast to winter
1. The upward EP flux and its convergence in the polar stratosphere are evident of their
contribution towards the weakening of the U50 and a general dominance over the effects of thermal
forcing by aerosols that have been, at this stage, substantially reduced (Fig. 1).
Similar wave propagation pattern as identified in the *cpl* experiment is known to be associated
with SSWs. We suspect that the decrease in the T50 difference between mid- and high latitudes
can act as a trigger for a weaker polar vortex in addition to the stratosphere absorbing the upward
propagating waves that is known to cause warming over the polar cap (Kodera et al., 2016;
Kretschmer et al., 2018). We will see further evidence of this in the next section.

**3.2.3. Third post-eruption winter**
The results in this section only refer to the *cpl* experiment since winter 3 is lacking in *atm-only*.
The SSW-like pattern of winter 2 clearly continues into winter 3, where most of the volcanic
aerosols have decreased to the extent that their radiative impacts no longer dominate. An exception
is the confinement of T50 warming over the polar stratosphere (Fig. 4c). Furthermore, anomalous
upward propagation of planetary waves continues to persist (Fig. 4f). This upward wave flux in
addition to the T50 warming resembles a pattern that behaves much like absorbing SSWs defined
by Kodera et al. (2016). To examine this response further we define SSWs based on the reversal
of the zonal-mean zonal winds at 60° N and at 10hPa between November and March according to
the method of Charlton and Polvani (2007) .

Results from the SSW analysis are presented in Fig. 6. No significant increase in SSWs is found
in winter 2, despite the SSW-like pattern detected. This changes in winter 3 when the difference
between perturbed and unperturbed experiment becomes statistically significant, with 27 SSWs
occurring in our forced experiment compared to only 6 in the control experiment. This increase in
SSWs agrees well with the U50 and T50 anomalies of winter 3 (Fig. 4c). During winter 2, the
warming of the polar stratosphere is as strong as in winter 3 but more spread out into midlatitudes.
These results are also in agreement with the stratospheric Plumb flux in winter 3 (Fig. S1c) where

 the upward flux is mostly circumpolar between 40° and 60° N, showing further evidence of the

 SSWs detected.

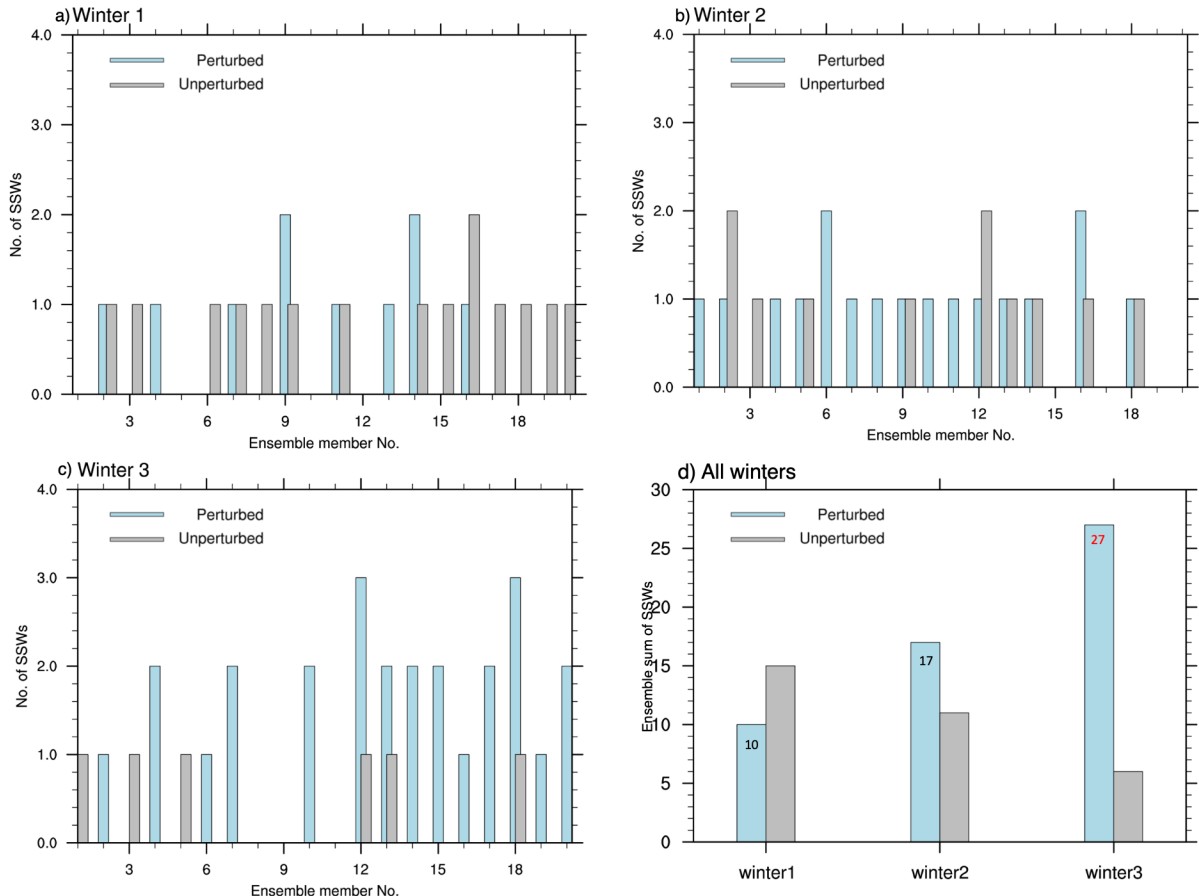

 Figure 6: a-c) The number of SSWs during winters 1-3 for each ensemble member in the *cpl*

 experiment and d) The sum of all SSWs in each experiment for all 20 ensemble members of winters

 1-3 both for *cpl* (light-blue bars) and control (gray bars). The color red indicates 95% significance

 according to a two-sided Student's t test.

 When comparing the ensemble sum of SSWs in the perturbed and unperturbed experiment using

 a Kolmogorov-Smirnov test (Fig. 6d), a significant increase in the number of SSWs occurs in

 winter 3 (p=0.0135). This underlines the generally strong SSW response occurring in winter 3,

 when the fraction of ensemble members having more than 1 SSW per winter increases to 50% (10

 ensemble members) in winter 3 compared to only 10% in winters 1-2. Of these 10 ensemble

 members, two members show three SSWs per winter that can be considered highly unlikely based

on historical records. Although winters with more than 1 SSWs are considered unusual, examples
do exist in the observational record of multiple SSWs in one winter, like the winter of 1998/1999
and 2009/2010 (Kodera et al., 2016 and Ineson et al., 2023 respectively).
To better understand the cpl SSWs response, we also did an SSW analysis on *atm-only* (Fig. S7)
where 50-75% less SSWs were detected in the perturbed simulation compared to the unperturbed
one. Such a response should not be unexpected during the forced polar vortex strengthening as
detected in *atm-only* (see Fig. 5). Furthermore, only single SSWs per winter were detected in all
20 ensemble members of the perturbed simulation while two (one) ensemble member(s) detected
double SSWs per winter 1 (winter 2) in the unperturbed simulation. Although these results do
suggest an increase in the number of SSWs in the *cpl* simulation, internal variability is large and
the frequency of SSWs fluctuates substantially between the three winters in the unperturbed
simulation.
We explored the impact of the ensemble size for the ensemble spread of two key diagnostics of
our mechanism, namely U10 and SSW, calculated as the standard deviation of post-eruption paired
anomalies for the first three post-eruption winters (Supplementary Fig. S8). Winter 3 produces
larger spread than winters 1 and 2, indicative of a least constrained forced response, which is
especially evident for ensemble sizes larger than 15. Therefore, only much larger ensembles may
provide signals not encompassing the value of zero within uncertainty.

According to the above, the evolution in *cpl* from winter 1 to winter 3 can be summarized as
follows: In the first winter, the thermal forcing appears to be stronger than the upward wave flux
because of the large amount of aerosols present, thereby dominating the response that causes the
polar vortex strengthening and the inclusion of cold polar air within. In the second winter, the
thermal forcing from the volcanic aerosols at midlatitudes has decreased where it is now mostly
confined to higher latitudes as seen both in the LW flux and T50 (Fig. 2f and Fig. 3b). We suspect
that in addition to the aerosol decrease, this slight decrease in the temperature difference between
high and midlatitudes allows the strong upward wave flux to dominate and enter the upper
stratosphere. There in the stratosphere, the waves are absorbed that causes further warming over
the polar cap in addition to weakening the zonal stratospheric winds (Fig. 5b and Fig. 4b). This
upward wave flux and weaker winds continue into the third winter, where winter 2 potentially acts
as a precursor, allowing for SSWs to develop more frequently as detected in the T50 warming that

is now confined over the polar cap (Fig. 4c and Fig. 5c, respectively). The SSW development is also evident in both U10 and U50 and T50 timeseries (Fig. 3c and Figure S3a-b respectively), where peak T50 warming occurs late in winter 3. The expected absence of a surface response is obvious in our *atm-only* experiment where basic physical mechanism, via the thermal wind balance due to radiative heating, dominates the atmospheric circulation response in the first two post-eruption winters. The strong stratospheric polar vortex then isolates the cold air over the polar regions (Fig. 5a-b) as is the case in winter 1 of in the *cpl* experiment (Fig. 4a).

### 3.3. Tropospheric response

What is it then that drives this polar vortex weakening and the SSW response in the *cpl* experiment? To examine in more detail the origin of the upward wave fluxes in winters 2 and 3 of the *cpl* experiment that causes the detected polar vortex weakening and the SSWs, we now turn our attention towards the troposphere.

We begin by comparing the response of T2m, vertical Plumb flux at 850 hPa and 200 hPa zonal wind, in *cpl* (Fig. 7) and in *atm-only* (Fig. 8).

As a response to the decrease in SW flux following the eruption, extensive and heterogeneous cooling is identified in the T2m anomalies in winters 1-3 (*1-2 for atm-only*) over latitude bands that contain the most significant SW flux decrease (Fig. 2d, Fig. 7a-c and Fig. 8a-b). The strongest cooling occurs over northeastern North America and along the Asian midlatitudes in winter 1, with much larger amplitude in *cpl* than in *atm-only* (Fig. 7a versus Fig. 8a). In *cpl*, a significant cooling is identified in the SST (Fig. S5), extending the area of negative T2m anomalies towards the ocean, in particular over the northwestern North Pacific in winter 1 (Fig. 7a). There it progresses from an initial preferential surface cooling over the midlatitudes in winter 1 to a later cooling of polar regions in winter 3 (Fig. 7c). In *atm-only*, the surface response is hampered over the ocean by the experimental design, and T2m anomalies are therefore confined to landmasses, yielding an overall much weaker temperature response compared to *cpl* (Fig. 8a-b).

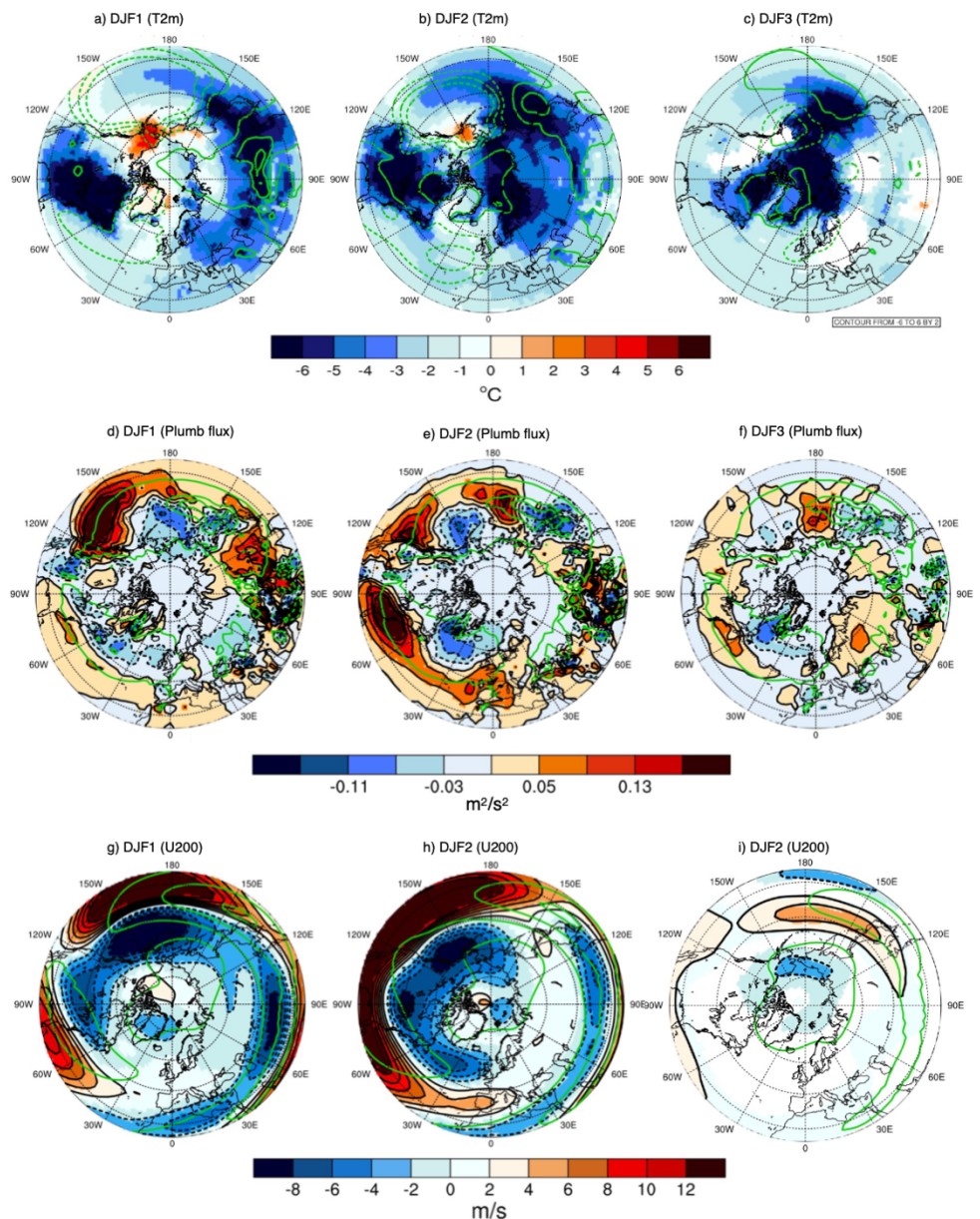

461

Figure 7. a-c) 2-meter air temperature (°C) in *cpl* (color) and sea-level pressure (green contours)
anomalies for winters 1-3. d-f) The vertical component of the Plumb flux (m²/s²) at 850 hPa in *cpl*,
and the climatology as green contours from -8 to 12 by 2, for winters 1-3. g-i) 200 hPa zonal wind
(m/s) anomalies in *cpl*, and the climatology as green contours from -0.15 to 0.15 by 0.04, for
winters 1-3. Contours and colored areas indicate significance at the 95% confidence interval
according to a Student's t-test.

468

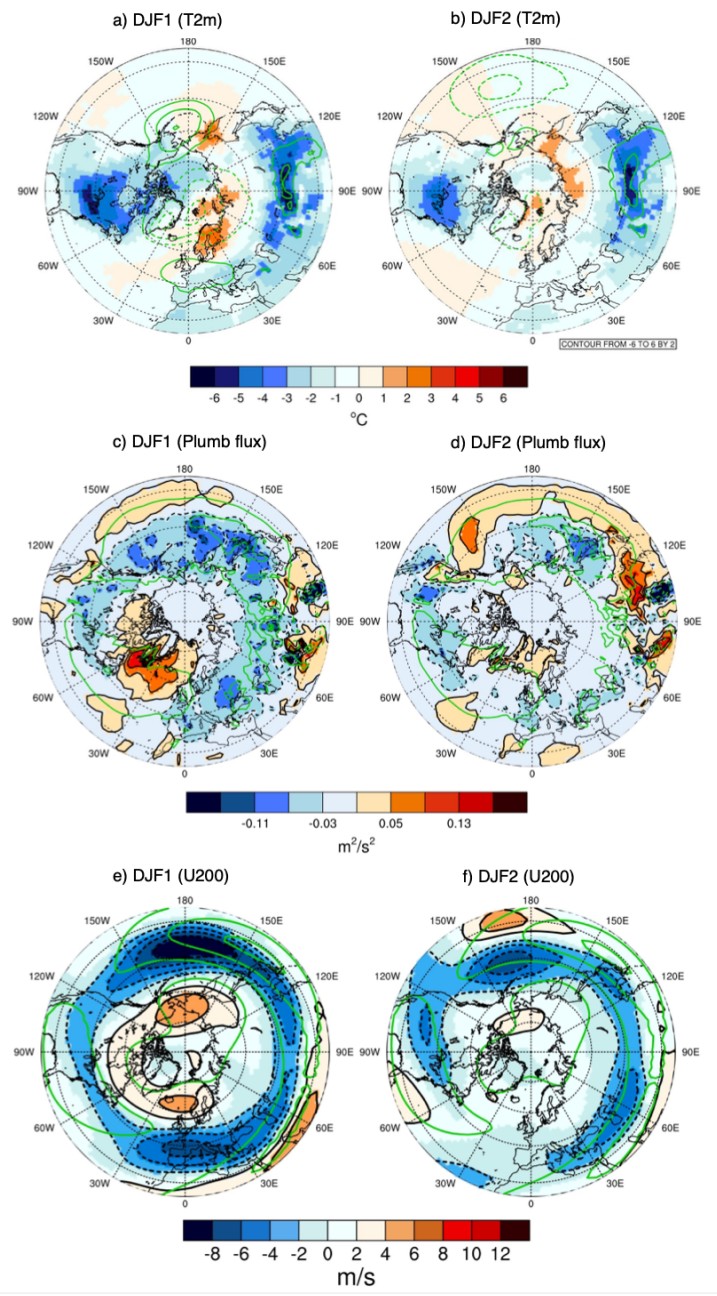

469

Figure 8. The same as Figure 8 but for the first two winters in atm-only. Contours and shaded areas

indicate significance at the 95% confidence interval according to a Student's t-test.

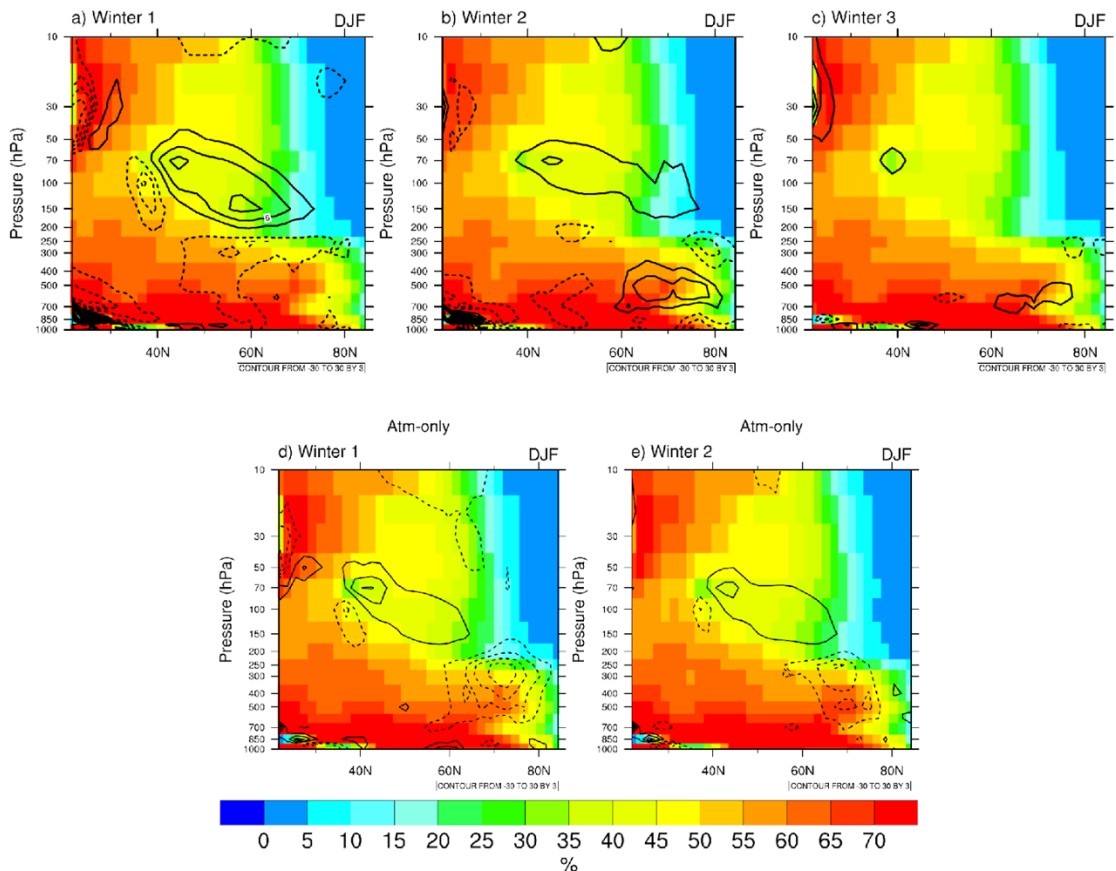

Figure 9. Probability (%) of favorable propagation conditions for large-scale stationary Rossby waves (zonal and meridional wavenumbers 1, 2 and 3) as a function of latitude and pressure levels (shading). Contours show the response in long_summer versus control. a) Winter 1 in *cpl*. b) Winter 2 in *cpl*. c) Winter 3 in *cpl*. d) Winter 1 in *atm-only*. e) Winter 2 in *atm-only*.

The vertical component of the the Plumb flux at 850 hPa (Fig. 7d-f) allows us to locate the origins of the upward EP flux in *cpl* (Fig. 4d-f). It is strongest over the north eastern part of the Pacific Ocean (off the west coast of North America) in winter 1, where it continues up into the lower stratosphere at 150 hPa (see Fig. S1a). In winter 2, the Plumb flux has decreased in the North Pacific and increased over the North Atlantic and Siberia, pointing to a possible influence of the change in land-sea temperature contrast (Fig. 7e). In addition to this upward flux, we also detect downward wave flux over both the Aleutian and Greenland regions at 850 hPa and over a large area south of 45° N at 150 hPa (Figure S1b). This downward Plumb flux is evidence of changes in the planetary wave structure where wave reflection occurs due to the sudden weakening of the zonal winds identified in the U10 (Fig. 3a). In winter 3, the Plumb flux now dominates both at 850

(seen in Fig. 7f) and 150 hPa (Fig. S1c), where it encircles the polar stratosphere north of 60° N.
In line with the weak EP-flux response shown in Fig. 5, the Plumb flux anomalies are generally
weak in *atm-only* compared to *cpl*, for both winter 1 and 2 (Fig. 8c-d).

Since upward wave activity depends on wave-mean flow interactions, several factors are at play
to explain the strong response in *cpl* vs *atm-only*. First, the change in zonal flow is substantially
different between the two pairs of experiments, as shown by the U200 anomalies (Fig. 7g-i and
Fig. 8e-f). In the first two winters we observe an intense deepening of the Aleutian low in *cpl* (Fig.
7a-b) associated with a large equatorward shift of the subtropical jet over the North Pacific (Fig.
7g-h, also seen in the zonal-mean averages of Fig. 4). The change in zonal flow is not as large in
*atm-only*, where there is a general decrease of U200 on the poleward side of the subtropical jets,
rather than a marked equatorward shift as in *cpl*. This further emphasizes that amplified surface
coupling when the ocean is coupled to the atmosphere has a dramatic impact on the amplitude of
the tropospheric response. Because the zonal flow acts as a waveguide for large-scale planetary
waves, we expect changes in upward wave propagation in the stratosphere. To measure how
waveguides change, we compute the probability of favorable propagation conditions for large-
scale stationary waves (Fig. 9). This is averaged for zonal wave numbers k=1,2,3 and meridional
wave numbers l=1,2,3, as a function of pressure and latitude (see section 2 for more details). Areas
of high probability show where large-scale waves preferentially propagate, while low probability
regions indicate where linear wave propagation is hampered. Generally, the mid-latitude
troposphere is more favorable for wave propagation than the high-latitudes and the stratosphere,
consistent with the tendency for stationary waves to propagate upwards and to be deflected towards
the equator, in climatology. After injection of the volcanic forcing, both *cpl* and *atm-only* exhibit
an increase in the probability for wave propagation between 40 and 60 °N in the lower stratosphere
during winter 1 and 2, but the responses in the troposphere are markedly different. In *atm-only*,
wave propagation is inhibited in the free troposphere north of 60°N, for both winters 1 and 2 (Fig.
9d-e), which is consistent with the EP-flux anomalies of Fig. 5. This response is absent from *cpl*
during winter 1 (Fig. 9a), and opposite during winter 2 when an increase of favorable conditions
for wave propagation is diagnosed (Fig. 9b). We also see that the waveguide has greatly reduced
in the subtropical troposphere in *cpl* winters 1-2 that favor large-scale waves to be redirected
towards the pole. This increase in favorable conditions for wave propagation in the troposphere
between 60 and 80 °N persists during winter 3 in *cpl* (Fig. 9c), which is a partial and the most
likely explanation for enhanced upward wave propagation in the stratosphere described in Fig.
4f.

In *cpl* winter 3, when the cooling is reduced in the NH mid-latitudes and has migrated towards the
polar regions (also evident in SST, see Fig. 5S), the amplitude of the 850 hPa upward Plumb flux
anomalies decreases compared to previous winters (Fig. 7f). This suggests that the mid-latitude
spatiotemporal cooling pattern plays a part in the strong wave activity detected near the surface.
This can be revealed by computing the T2m gradient (Tgrad) where strong land-sea temperature
gradients are known for their ability to influence atmospheric wave activity (Hoskins and Valdes,
1990; Brayshaw et al., 2009; He et al., 2014; Wake et al., 2014; Portal et al., 2022). Figure 10a
shows sharp significant changes in the meridional gradient that encircles 45° N in winter 1, with
positive (negative) gradient anomalies occurring south (north) of 45° N. In winter 2 we still see
the gradient present at 45° N but now located over North America and the North Pacific. Winter 3
mostly reveals regional anomalies in the Barents-Kara, Greenland-Iceland and the North Pacific
regions (Figure 10b-c), occurring over  areas of significant sea ice increase (not shown). The sharp
Tgrad change in winter 1 (Fig. 10a) is followed by a reduction of land-sea temperature contrast
over eastern Canada and the U.S. in winter 2 (Fig. 10b). This is a known cause of planetary wave
enhancement (Portal et al., 2022) and could provide an explanation for the strong surface upward
wave flux detected in the second and third post-volcanic winters (Figure 7e-f). Both the zonal and
meridional Tgrad components show an increase in the northern part of Alaska that coincides with
the region of T2m warming over the Aleutian/Alaska region (Fig. 7a) and the strong upward Plumb
flux (Fig. 7d). This warming, in addition to the strong continental cooling over North East America
and the general decrease in Tgrad spanning from mid to northern part of North America, might
influence this strong Plumb flux anomaly in the North Pacific. Of note, sea-ice extent increases
around East Siberia extending into the Chukchi Sea (not shown), highlighting the potential
influence of sea ice variability on Tgrad and upward Plumb flux anomalies in the area.
Plotting the average Tgrad for various regions against the average number of SSWs for winters 1-
3 (Fig. S6), we do see that the strongest Tgrad reduction occurs over the North East US in the
second winter. This is in agreement with the upward Plumb flux over the same region and serving
as further evidence for its contribution to the upward EP flux in winter 2. This Tgrad reduction
continues into the third winter where we also detected a reduction in the upward Plumb flux over
the same area (Fig. 7f). Looking towards the Barents Sea, a clear spatial difference emerges
compared to the North East US, where a clear Tgrad increase occurs in winter 3 related to the
SSWs. In general less changes are detected between winters in the North West NA and the North
Pacific, reflecting the confined cooling over higher latitudes in winter 3 associated with the SSWs.

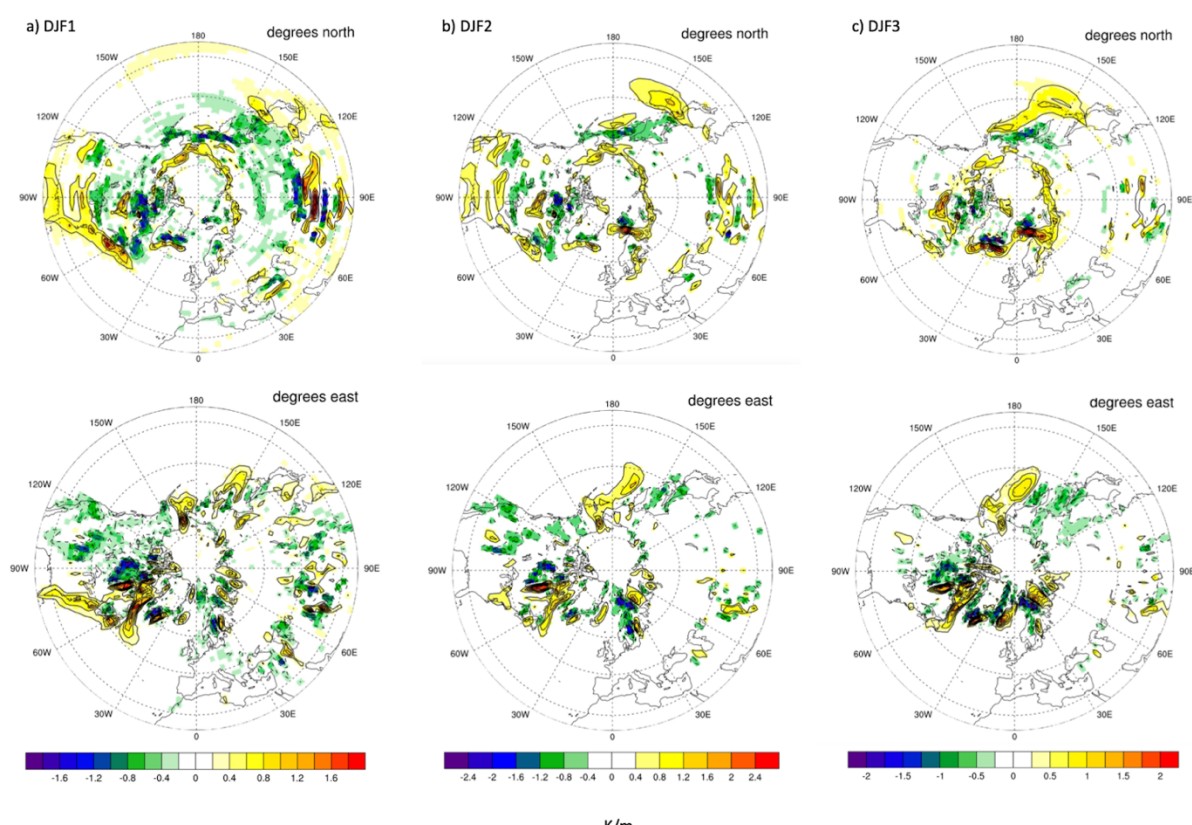


Figure 10: a-c) The zonal (degrees north) and meridional (degrees east) T2m gradient anomalies
(perturbed minus unperturbed) for winters 1-3. Contours indicate 95% significance according to a
Student's t-test. Note the different colorbar for each winter.

To complete our assessment of the tropospheric response, we examine if eddies play a role in the
*cpl* polar vortex response and the anomalous upward EP flux (Fig. 4a-f) as well as SSWs detected
in winter 3 by following Smith et al. (2022) (see Methods). This is done for both perturbed (red)
and unperturbed (blue) experiments in *cpl* and *atm-only* (Fig. S4). We see an increase in perturbed
eddy feedback at around 40-70° N both for *cpl* winter 1 and winters 1-2 in *atm-only* during the
polar vortex strengthening (Fig. S4). However, the role of eddies in the polar vortex weakening in
winters 2-3 is unclear, especially considering the eddy feedback increase of the control run in
winter 2 (Fig. S4b). In general, these results suggest that the signal-to-noise ratio is too small to
identify a role for eddy feedback in our experiments.

**4 Discussion**
Our two sets of coupled ocean-atmosphere (*cpl*) and atmosphere-only (*atm-only*) experiments
examine the large-scale climate response to an idealized long-lasting NH eruption, where their
differences give us valuable insight into the volcanically forced mechanisms at play within the
coupled climate system in CESM1. Specifically, we analyzed the first three winters of the *cpl*
experiment and used the first two winters of *atm-only* as a comparison to investigate the dynamics
that govern the post-eruption stratospheric polar vortex and the associated surface response.
Results from the *cpl* experiment show a similar response in the first winter as in the two winters
of *atm-only*, with a strengthening of the zonal winds resulting from an aerosol-induced sharp
temperature gradient between the mid-latitudes and the pole (Fig. 4a and Fig. 5a). We show that
this zonal wind strengthening is not affected by the detected strong upward EP flux, where the LW
flux (Fig. 2e-f) supports our conclusions that the polar vortex strengthening is induced by the
thermal wind balance. A distinct change to this pattern emerges in *cpl* winter 2 where we detect an
SSW-like pattern, with strong negative anomalies emerging in the polar U50 winds and a warming
in the T50 field (Fig. 4b). We also detect an LW absorption at high latitudes, that is absent at
midlatitudes, where this T50 warming is evident of the potential role of a decreased temperature
gradient in the identified polar vortex weakening. Furthermore, the upward wave-activity flux
from the troposphere into the stratosphere and the T50 warming indicate absorption of upward
propagating waves into the stratosphere that causes this warming and weakening of U10 winds
over the polar cap. This pattern is known to be related to SSWs (Kretschmer et al., 2018; Kodera
et al., 2016), agreeing with our results in winter 3 where an increase in the occurrence of SSWs is
detected. The strong upward EP flux greatly depends on ocean-atmosphere coupling originating in
the surface cooling in addition to the changes in upper tropospheric zonal flow. This further
contributes to the upward EP flux from the troposphere into the stratosphere that eventually leads
to polar vortex weakening and enhanced SSWs.
Although the above coincides with a positive (negative) eddy feedback in the first (second) winter
that could in theory play a role in sustaining the strengthening (weakening) of the polar vortex, our

eddy feedback results indicate low signal-to-noise ratio where further studies with additional ensemble members would be required to confirm their role in the forced response. We also note that Smith et al. (2022) identified CESM1 WACCM-SC as having one of the weakest eddy feedback of the sixteen models they investigated, so the response of eddy feedback may be more significant in other models. Similar to the eddy feedback, low signal-to-noise ratio is also evident in the SSW analysis. However, the response we detect in the U50 and T50 fields is strong compared to the unperturbed simulation where the SSWs provide an explanation in agreement with the patterns detected. Furthermore, the SSW analysis for *atm-only* and the ensemble size test (Fig. S7 and Fig. S8 respectively) both show strong evidence of a robust signal for winter 3 despite the noisy polar vortex and the limited ensemble size. We also see that the large decrease in SSWs in the perturbed simulation of *atm-only* (when compared to unperturbed) is consistent with the detected polar vortex strengthening. This further supports the significance of the signal we detect in *cpl* winter 3 compared to the background noise. In addition, all winters examined, in both *cpl* and *atm-only*, showed that there is up to 15% chance of getting more than 1 SSWs per winter in all ensemble members. This is not far from Ineson et al. (2022) who identified a double event once every 9 years in a 66-year ERA5 record. The exception is *cpl* winter 3 that is also the only winter that has 3 SSWs, with the average SSW occurrence also being the only winter above 1 (1.17) while all other winters span between 0.15-0.85 per winter. A similar NH high-latitude eruption has not taken place during the observational period, so we have no comparison. Also, to the best of our knowledge, a similar high-latitude sulfur injection study has not been performed before. Therefore, it is difficult to say at this stage if such a response is realistic or not, but in general more than two SSWs per winter can be considered exceptional yet plausible, as is also the case for our idealized eruption.

Bittner et al. (2016b) identified an opposite response driven by a similar underlying mechanism, when compared to our *cpl* response in winters 2-3 (Fig. 4), following a Tambora-like eruption where a strengthening of the polar vortex due to less wave breaking at high latitudes was considered to be an indirect effect associated with a changes in planetary wave propagation. Since the volcanic aerosols in our experiments have declined extensively in the third winter, making the aerosol thermal forcing a limited factor, we cannot rule out similar indirect effects where changes

in wave propagation leads to an increase in wave breaking at high latitudes and hence the increase
in SSWs.
While not directly comparable to our study but still providing an important analog, Muthers et al.
(2016) identified an average increase in the number of SSWs during a 30-year (constant) decrease
in solar radiation in line with our significant increase in SSWs in winter 3. Our results do support
the findings of Sjolte et al. (2019), where the stratospheric temperature gradient does not appear
to play a major role in the polar vortex weakening we identify, while the upward wave flux does.
The strong surface cooling detected in Fig. 7 is a well-known caveat in CMIP5 models (including
CESM1) (Driscoll et al., 2012; Chylek et al., 2020) and is clearly detected in our coupled
simulations. Since our results indicate the dominant role of the volcanically induced stratospheric
thermal wind response that causes the polar vortex strengthening, the cooling does not appear to
impact the response identified in winter 1. This is also revealed by the EP flux. The same cannot
be said about winters 2-3, where our results indicate that the exaggerated spatiotemporal T2m
pattern might explain the strong upward wave flux and the associated stratospheric response.
Interestingly, a slight difference between *atm-only* and *cpl* is detected in both the LW and SW flux
that is caused by a strong significant decrease in high cloud cover in the *cpl* simulation (not shown).
This cloud cover decrease, especially at mid- to high latitudes, agrees with the increased LW fluxes
at higher latitudes in addition to the decrease in SW flux and the associated surface cooling. This
raises a question regarding the role of forced surface processes in these high cloud changes, which
we leave open for further studies.

As mentioned in the methods section, we assume a similar lifetime of volcanic aerosols at 65° N
as at 45° N. When considering the e-folding time in Toohey et al. (2019), a substantial aerosol
decrease of about 43% occurs at 17km (a.b.s.) for an eruption at 60° compared to at 0°. However,
since our experiment assumes a constant stratospheric injection over 5 months with the aim to
simulate a long-lasting HL eruption compared to a single injection at low latitudes, the difference
in the e-folding time between low and high-latitudes would be expected to decrease. Using
CESM2-WACCM6 with interactive chemistry Zhuo et al. (2023) identified that although an
eruption at 64° N did have a shorter aerosol lifetime compared to one at 15° N, it leads to stronger
volcanic forcing over the NH extratropics. In addition, one of their conclusions was that different
duration and intensity of both tropical and NH extratropical eruptions can lead to different results,
stressing that our 6 month long stratospheric sulfate aerosol enhancement is not directly
comparable with volcanic eruptions of shorter duration. Although the aerosol lifetime in our
experiment might be exaggerated into the third year, our results do indicate that the polar vortex
weakening in winter 2 appears to act as a trigger for further weakening that eventually leads to
SSWs in winter 3. In order to increase confidence on such a delayed link, additional sensitivity
simulations are required, which we we leave that for future studies.

Unlike our eruption simulated using a version of WACCM4, where the chemistry is prescribed,
natural volcanic eruptions can contain various chemical compounds that impact the formation and
the lifetime of sulfate aerosols as well as affect the atmospheric circulation via, e.g., ozone
depletion, like halogens are known to do. More advanced versions as well as models that include
interactive chemistry are thus important to reveal in more detail the chemistry-climate interactions
that occur in the natural world (Clyne et al., 2021; Case et al., 2023; Fuglestvedt et al., 2024). Thus
our idealized experiment can be considered primitive in the sense that it only considers sulfate
aerosols but sufficient when focusing on answering questions on the basic mechanism that such
eruptions can initiate. Another important aspect that we do not focus on in our study is the role of
different initial conditions on the forced climate response, where initial atmospheric and climate
conditions, including e.g. the stability of the polar vortex, control the lifetime and distribution of
the volcanic aerosols as well as the forced dynamic climate response (Zanchettin et al., 2019;
Weierbach et al., 2023; Zhuo et al., 2023; Fuglestvedt et al., 2024). An exception is our assessment
on how the easterly and westerly phase of the QBO affect our results where we compared ensemble
members showing easterly phase with the westerly ones to test if the U50 and T50 response
patterns would be different. They were not: both phases showed a weakening of the U50 although
the zonal winds were more confined and consistent over the higher latitudes of the NH during the
easterly phase (not shown). The difference in the number of ensemble members used for these
calculations could of course impact the statistics of this test of ours but not the overall pattern
detected.
CESM2-WACCM6 has obvious improvements when compared to CESM1-WACCM4 (see e.g.
Gettleman et al., 2019, Danabasoglu et al., 2020), among them being an interactive QBO as well
as having a slightly higher frequency of SSW occurrence (Holland et al., 2024). Nonetheless,
CESM1-WACCM4 has comparable transient climate response to CESM2 as well as the ability to
capture the general physical mechanism occurring within the climate system as identified in
various recent studies (Danabasoglu et al., 2020; Zang et al., 2018; Elsbury et al., 2021b; Peings
et al., 2023; Ding et al., 2023; Yu et al., 2024).

**5 Conclusions**

Through comparison of the *cpl* and *atm-only* simulations, our results clearly demonstrates the
important role of ocean-atmosphere coupling in the stratospheric response to enhancements of the
stratospheric sulfate aerosol layer at higher NH latitudes. We see that this aerosol enhancement
layer triggers two competing mechanisms in the first three winters:
i) Winter 1: The stratospheric polar vortex strengthening is triggered by stratospheric aerosol
thermal forcing via thermal wind balance. This response is not influenced by the strong upward
wave flux identified, originating in the forced surface cooling and changes in tropospheric
circulation, and provides strong evidence of two mechanisms that are competing simultaneously:
A Top-down and a Bottom-up mechanism, where the Top-down mechanism dominates the
response.
ii) Winter 2: The upward wave flux is absorbed in the stratosphere that causes a warming over the
polar cap and a polar vortex weakening. This pattern is similar to SSWs although its occurrence is
not significant. Here the Bottom-up mechanism dominates.
iii) Winter 3: The persistence of the upward wave flux continuing into the third winter leads to an
increase in SSWs with warming now confined over the polar cap, again demonstrating the
dominating Bottom-up mechanism as in winter 2.
It is clear from our results that the strong surface cooling following the HL sulphate aerosol
injection causes dramatic changes in tropospheric circulation. These changes further modify
atmospheric waveguides where we detect an increase in propagation of planetary waves in the
lower stratosphere occurring at higher latitudes. Although we do find similarities in the eddy
feedback when compared to the general climate signal that we identify, such as the decrease in
eddy feedback in winter 1 potentially sustaining the polar vortex strengthening, we emphasize its
weak signal. At the same time we encourage further studies on this subject, especially concerning
the lack of published comparison studies regarding both high and low latitude volcanic eruptions
and SSWs. Ideally such studies would include the latest model generations in addition to
observational datasets. They should also consider the impact of different climate realizations and
the eruption magnitude on the forced response. Furthermore, these results highlight the importance
of including high-latitude volcanic forcing simulations of various lengths and/or magnitudes in
projects such as VolMIP, especially considering the current volcanic unrest and increased activity
in some of the major volcanic systems in Iceland.

**Data availability**
The model output is available upon request by contacting the corresponding author.

**Author contribution**
HG conceptualized this study along with GM, YP and DZ. *cpl* and *atm-only* experiments were
carried out by HG and YP. Analysis and calculations of model output as well as graphical
representation was done by HG except for the eddy feedback and the probability of favorable
propagation conditions that was done by YP and the ensemble size test that was done by DZ.
Manuscript draft was done by HG and editing was done by DZ and YP. GM served as the principal
investigator of this work and did the final editing.

**Competing interests**
The corresponding author declares that none of the authors have any competing interest.

**Acknowledgement**
This work is supported by the Icelandic Research Fund (IRF), grant No. 2008-0445. HG
acknowledges the Fulbright Scholar Program, which is sponsored by the U.S. department of state
and Fulbright Iceland, that facilitated the stay of HG and her family in Irvine, CA during this work.
We also acknowledge high-performance computing support from Cheyenne
(doi:10.5065/D6RX99HX) provided by NCAR's Computational and Information Systems
Laboratory, sponsored by the National Science Foundation that we used for our experiments. HG
wants to thank the staff at the department of Earth System Science at UCI for the facility and
assistance during HG's stay at UCI. Finally, we also want to thank Matthew Toohey for his
assistance in the interpolation of the forcing files for WACCM4.

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
