# Peer review of "Stratospheric circulation response to large Northern high-latitude volcanic eruptions in a global"

_EGUsphere, 2024_

## Author Comment (AC1)

**Response by authors**

We thank Reviewer #1 for their appreciation of our study and for their helpful comments on the discussion paper.

We are convinced that a revision of our original manuscript that accounts for the Reviewer's criticism is feasible. Specifically, to address the four major points raised by the Reviewer, in the revised manuscript we plan to:

- Better put CESM1-WACCM in the context of state-of-the-art chemistry climate models (see also our response below), also referring to recent studies using the same model version used here (e.g., Jiang et al., 2024; Li et al., 2023; Clyne et al., 2021).
- Consider rephrasing critical aspects of the manuscript (such as the title and the abstract) to de-emphasize our experiments as volcanic eruption ones. A possibility we are considering is to refer to stratospheric aerosol injection experiments, instead. This should also solve other questions about the comparability of our results with observations from historical events raised by the Reviewer in their specific comments;
- Provide additional information about the spatio-temporal characteristics of volcanic aerosol and associated radiative flux anomalies;
- Better illustrate and discuss the spatial structure of anomalies involved in the bottom-up mechanism, in particular better clarify the role of the land-sea temperature contrast in the North Pacific/Western North American region, which appears to be at the core of the wave activity anomaly.

More specifically on the realism of the model version used here with respect to current versions, focusing on the atmospheric component WACCM, we will refer to Gettleman et al. (2019), who provide an extensive comparative analysis of versions 4 and 6, with a detailed description of biases in both versions. There are obvious improvements in WACCM6 compared to WACCM4, such as a higher resolution, an interactive QBO (nudged in our simulations) and more evolved chemistry. One of the important aspects of atmospheric variability for our study is the simulation of NH sudden stratospheric warming events, for which WACCM6 has a slightly higher frequency of occurrence than WACCM4 (mostly due to more late winter simulated events). We will also refer to papers comparing general climatic features of CESM1 and CESM2 (e.g., Mills et al., 2020; Holland et al., 2024): as typically found also for other models, the papers point to the critical role of simulated cloud responses and feedbacks for explaining differences across results from different model generations.

The Reviewer also expresses some concerns on the stratospheric cooling over the polar cap detected in the first winter in both the cpl and atm-only experiments. We do acknowledge that this needs to be more clearly presented and discussed. In brief, during boreal winter there is no aerosol-induced stratospheric warming at high latitudes, which is why the warming is only detected at mid-latitudes where the aerosol still has an impact on shortwave radiation. As we mentioned in the original submission, the polar vortex strengthening then encloses cold polar air at high latitudes that explains the

stratospheric cooling identified over the polar cap. In addition to better describing this mechanism in the revised manuscript we will add more analysis to support our interpretation.

The last major comment of Reviewer #1 concerns the aerosol induced stratospheric warming where according to him it should be weakened in the second and third winter due to decreased aerosol concentrations that in turn could cause the detected polar vortex weakening in the 2nd and 3rd winter. This is a valid point. However, in order to address this concern we would at least need an additional cpl experiment where the aerosol concentration is held at constant maximum values over, say, 2-3 years before declining. Indeed that would be of interest but out of the current scope. Thus we will leave it for future studies and in stead add discussions on this possibility in a revised manuscript.

The concerns that Reviewer #1 has on the definition and number of SSWs are closely related to the concerns of Reviewer #2. We acknowledge that we need to expand the definition of SSWs that we use in the identification algorithm based on Charlton and Perez, 2007 and detailed therein. Regarding the number of SSWs we refer to Figure 1R and the associated answer in our response to Reviewer #2.

Furthermore, all the specific comments by the Reviewer are pertinent, and we will account for all of the suggestions and requests of clarification in the revised manuscript. In particular, also following the suggestion by Reviewer #2 to restructure the results section of the paper, in the revised manuscript we plan to articulate it as follows: first, present the whole dynamics in the cpl experiment for the three post-eruption winters, in sequence; then, present the relevant dynamics in the atm-only experiment to disentangle the role of the bottom-up mechanism in the second post-eruption winter only. We will also largely rewrite the discussion. We plan to move most parts about the dynamical interpretation to the results section (also following a comment in this direction by Reviewer #2), to refocus the discussion on limitations of our study and on broader implications for our general understanding of aerosol-climate interactions. All statements will also be checked for correctness and clarity, tackling all the technical comments by the Reviewer.

**References**

Jiang, J., Xia, Y., Cao, L., Kravitz, B.,MacMartin, D. G., Fu, J., & Jiang, G.(2024). Different strategies of stratospheric aerosol injection would significantly affect climate extreme mitigation. Earth'sFuture, 12, e2023EF004364

Lee, W. R., Visioni, D., Bednarz, E. M.,MacMartin, D. G., Kravitz, B., & Tilmes,S. (2023). Quantifying the efficiency of stratospheric aerosol geoengineering at different altitudes. Geophysical ResearchLetters, 50, e2023GL104417

Clyne, M., Lamarque, J.-F., Mills, M. J., Khodri, M., Ball, W., Bekki, S., Dhomse, S. S., Lebas, N., Mann, G., Marshall, L., Niemeier, U., Poulain, V., Robock, A., Rozanov, E., Schmidt, A., Stenke, A., Sukhodolov, T., Timmreck, C., Toohey, M., Tummon, F., Zanchettin, D., Zhu, Y., and Toon, O. B.: Model physics and chemistry causing intermodel disagreement within the VolMIP-Tambora Interactive Stratospheric Aerosol ensemble, Atmos. Chem. Phys., 21, 3317–3343, https://doi.org/10.5194/acp-21-3317-2021, 2021.

Holland, M. M., Hannay, C., Fasullo, J., Jahn, A., Kay, J. E., Mills, M., Simpson, I. R., Wieder, W., Lawrence, P., Kluzek, E., and Bailey, D.: New model ensemble reveals how forcing uncertainty and model structure alter climate simulated across CMIP generations of the Community Earth System Model, Geosci. Model Dev., 17, 1585–1602, https://doi.org/10.5194/gmd-17-1585-2024, 2024.

Danabasoglu, G., Lamarque, J.-F., Bacmeister, J., Bailey, D. A., DuVivier, A. K., Edwards, J., et al. (2020). The Community Earth System Model Version 2 (CESM2). Journal of Advances in Modeling Earth Systems, 12, e2019MS001916. https://doi.org/10.1029/2019MS001916

Gettelman, A., Mills, M. J., Kinnison, D. E., Garcia, R. R., Smith, A. K., Marsh, D. R., et al. (2019). The whole atmosphere community climate model version 6 (WACCM6). Journal of Geophysical Research: Atmospheres, 124, 12380–12403. https://doi.org/10.1029/2019JD030943

Mills, M. J., Schmidt, A., Easter, R., Solomon, S., Kinnison, D. E., Ghan, S. J., Neely III, R. R., Marsh, D. R., Conley, A., Bardeen, C. G., and Gettelman, A.: Global volcanic aerosol properties derived from emissions, 1990–2014, using CESM1(WACCM), J. Geophys. Res.-Atmos., 121, 2332–2348, https://doi.org/10.1002/2015JD024290, 2016.

Mills, M. J., Richter, J. H., Tilmes, S., Kravitz, B., MacMartin, D. G., Glanville, A. A., Tribbia, J. J., Lamarque, J-F., Vitt, F., Schmidt, A., Gettelman, A., Hannay, C., Bacmeister, J. T., and Kinnison, D. E.: Radiative and chemical response to interactive stratospheric sulfate aerosols in fully coupled CESM1 (WACCM), J. Geophys. Res.-Atmos., 122, 13061–13078, https://doi.org/10.1002/2017JD027006, 2017.

---

## Author Comment (AC2)

**Response by authors**

We thank Reviewer #2 for their appreciation of our study and for their helpful comments on the discussion paper. We are convinced that a revision of our manuscript that addresses the Reviewer's criticism on methodological and presentation aspects of our study is feasible. Specifically, we plan to revise the manuscript as outlined in the following.

Concerning the Reviewer's skepticism about the substantial increase in SSWs in the third post-eruption winter, in the revised manuscript we will support our analysis with an additional figure that illustrates the number of SSW events in individual realizations and an improvement of original Figure 3 showing a time series of the stratospheric wind for individual realizations. The figure shown below (Figure 1R) provides a glimpse on the distribution of SSW events in individual realizations (x-axis) and for the first four post-eruption winters (DJF1-4), for the cpl and control experiments. The figure shows that whereas the fraction of cpl realizations with at least one SSW event in winter 3 is the same as in winter 2 (70%), the fraction of cpl realizations with more than one SSW event in winter 3 (50%, so half of the realizations) is unmatched in other winters, both in cpl and control experiments. In the revised manuscript we will support this comparative analysis with statistical tests (ranksum).

a)

[Figure]

b)

[Figure]

c)

[Figure]

Figure 1R: *a)* The number of SSW occurrences (y axis) during the first 4 post-volcanic winters (x axis) for each of the ensemble members of the cpl experiment. The fraction of ensemble members having more than 1 SSW per winter increases to 50% (10 ensemble members) in winter 3 compared to only 10% (2 ensemble members) for winters 1 and 2. Winter 4 is shown only as an additional comparison. *b)* The same as *a)* but for the control experiment. c) Anomalies (cpl minus control) of the stratospheric zonal winds at 50hPa (U50) at 70-80° N for each ensemble member (colored lines) and the ensemble mean (black curve). A more consistent U50 decrease is detected in winter 3 compared to winter 2 where a single realization causes a sharp drop in U50 (light-blue line).

In the revised manuscript, we will also better describe the mechanism responsible for the enhancement of SSW occurrence in the third post-eruption winter. Our focus will be on surface anomalies in the North Pacific/Western North American sector, where anomalous land-sea temperature gradients are suggested to contribute to the strong upward wave flux detected.

We plan to address the issue on the different aerosol lifetime for high-latitude eruptions compared to tropical ones in the revised version by adding a clarification on this matter in the Methods and Discussions chapters and compare our forcing with other related studies on high-latitude eruptions/aerosol injections. However, we do want to stress here that our experiment is designed around a longer-lasting aerosol injection, unlike the experiments in Toohey et al. (2019), with maximum values spanning in total 5 months that could potentially counteract part of this decrease.

We also appreciate the comment about improving the EP and Plumb flux analysis, which is central to our study. In a revised version of the manuscript we will better connect to existing literature and cite all the references suggested by the Reviewer. Especially we will dig deeper into the mechanism by, e.g., adding an analysis to assess a) the role of the eddy feedback parameter in the increased upward EP flux at the surface and b) the role of wave-eddy interactions regarding our detected stratospheric circulation/polar vortex response (preliminary results also point to a potential role of land-sea surface temperature gradients in the North Pacific/North American sector). As

a result, we will improve Discussions with respect to the above comment and take into consideration putting our results in context with DallaSanta et al. (2019) and/or Bittner et al. (2016) as the Reviewer mentions, both of which are highly relevant (see also Smith et al. 2022).

Finally, and in light of the comments from both Reviewer #1 and #2, we agree on altering the title to better represent and describe the content of this manuscript as well as to restructure the results section of the paper (see Author response to Reviewer #1 for details). Regarding the specific comments by Reviewer #2, in the revised manuscript we will account for each of the suggestions and the clarification requests.

References

Smith, D. M., Eade, R., Andrews, M. B., Ayres, H., Clark, A., Chripko, S., ... & Walsh, A. (2022). Robust but weak winter atmospheric circulation response to future Arctic sea ice loss. *Nature communications*, *13*(1), 727.

---

## Author Response (AR1)

Reviewer #1

We want to thank reviewer #1 (RW1) for the relevant and constructive comments he provided us with that we feel has contributed to the general improvements of this study. Here below we have addressed all of the comments made by RW1 where our responses are detailed in red. Since we have made considerable changes within the manuscript, part of the more detailed comments made by RW1 might no longer be relevant but in those cases we will refer to the changes made when appropriate.

**General comments:**
This study uses CESM1-WACCM4 for an idealized experiment with six-months volcanic aerosol injection in the Northern Hemisphere high latitude stratosphere. The authors emphasizes that potential increased sudden stratosphere warming events are found in response to the eruption. If the results are solid, then it can add valuable new insight to the research field. However, current experiment design, results and discussions in the paper are not convincing enough to make solid conclusions as stated in the paper.
Below are some major questions that need to be clarified/addressed:
· The study uses CESM1-WACCM4 with specified chemistry, which is already an old version of the model, how this old version is suitable for this study is not convincingly stated in the manuscript.

Indeed that is true, we have updated the Methods chapter (Pg 5, L125-131):
"We use the specified chemistry version of WACCM4 (SC-WACCM4), which is computationally less expensive to run, but simulates dynamical stratosphere-troposphere coupling and stratospheric variability like SSWs and the polar vortex with skills comparable to the interactive chemistry model version (Smith et al., 2014). CESM1/WACCM4 uses the Community Atmospheric Model Radiative Transfer (CAMRT) to parameterize the radiative forcing where it has been shown to accurately represent stratospheric aerosols by f. ex. simulating the temperature response following Mt. Pinatubo in 1991 (Neely et al., 2016)."

as well as added the following to the Discussion chapter (Pg 27, L641-647):
"CESM2-WACCM6 has obvious improvements when compared to CESM1-WACCM4 (see e.g. Gettleman et al., 2019, Danabasoglu et al., 2020), among them being an interactive QBO as well as having a slightly higher frequency of occurring SSWs (e.g., Mills et al., 2020; Holland et al., 2024). Nonetheless, CESM1-WACCM4 has comparable transient climate response to CESM2 as well as the ability to capture the general physical mechanism occurring within the climate system as identified in various recent studies (Danabasoglu et al., 2020; Zang et al., 2018; Elsbury et al., 2021b; Peings et al., 2023; Ding et al., 2023; Yu et al., 2024)."

· The experiment design, why the injection mass maintains over 6 months without any change and over wide vertical range (10-27 km), is it possible for a volcanic eruption? This looks more like a stratospheric aerosol modification case instead of a volcanic eruption case.

This is a valid point and we agree on changing the wordings in parts of the manuscript. We have also added the following explanation to this in the Methods section (Pg 6, L161-168): "In this experiment we assume stratospheric injection only, although similar eruptions in the natural world would likely inject part of the total aerosol mass within the troposphere during the eruption. Past NH eruptions like Laki and Eldgjá had an atmospheric loading of 210Tg and 120Tg respectively, much larger than our 14Tg eruption, that was carried aloft with the eruptive column up into the upper troposphere with portions of the aerosols reaching the lower stratosphere during the eruptions (Thordarson et al., 2001). Hence our experiment can also be considered as a 6-month stratospheric aerosol injection that is analogous to similar although smaller eruptions without the tropospheric aerosols."

· How is the aerosol distribution? What changes do aerosol zonal mean have along the time? How are the shortwave and longwave radiation and stratospheric temperature changes related to the aerosol distributions? These are all unclear questions, which make it hard to understand the mechanism explanation in the following parts.

We thank the reviewer for raising this important issue and we have now added additional figure panels in Figure 2 that do indeed act as a support to our results (see Pg 10).

· The manuscript emphasizes two competing mechanisms for explaining the results, one is the local stratospheric aerosol induced warming mechanism, the other one is the local surface cooling induced wave activity change mechanism, however, no results or supplement figures prove if the warming is at the same location as the aerosol loading area, or the increased wave propagation originates from the cooling areas.

We are certain that the new figure panels mentioned above in addition to Figures 3-5 are in support of our results and should provide the reviewer with sufficient arguments needed in support of the two mechanisms mentioned.

Besides, why there is a cooling in the first winter in both the cpl and atm-only runs are not well explained. In the winter, there is no shortwave radiation, then the aerosol induced stratospheric warming due to longwave radiation absorption should be a dominating effect compared to the first summer, but why it's the opposite? Connection to strengthened polar vortex in the winter needs to be explained better.

Again we refer to Figure 2, where the longwave flux in cpl and atm-only shows that warming from the aerosols is stronger at mid latitudes compared to at the poles throughout almost all of these three years in the cpl run but less so in atm-only. This initiates the thermal wind response that leads to the cooling. We have also added eddy feedback (see subchapter 3.3.1, Pg 18-19) assessment that shows weak evidence of their role in the polar vortex strengthening, occurring in both runs despite the absence of this LW difference between mid and high latitudes during winter (DJF). We have edited our Results and Discussions/Conclusions according to these additional figures and analysis.

Moreover, the aerosol induced stratospheric warming should be weakened in the second and third winter due to decreased aerosol, thus the strengthened polar vortex in the first winter can also be weakened due to less aerosol in the second and third winter, how can you rule this out from your second mechanism?

Another valid point. This is something we cannot rule out except with additional simulations that are not possible at this stage. However, we have added the following statement since it relates to the reviewer's comment as well as offering a potential explanation for our response in the third winter (Pg 25, L584-590):
"Bittner et al. (2016b) identified a similar but opposite response, when compared to our *cpl* response in winters 2-3 (Fig. 4), following a Tambora-like eruption where a strengthening of the polar vortex due to less wave breaking at high latitudes was considered to be an indirect effect associated with a changes in planetary wave propagation. Since the volcanic aerosols in our experiments have declined extensively in the third winter, making the aerosol thermal forcing a limited factor, we cannot rule out similar indirect effects where changes in wave propagation leads to an increase in wave breaking at high latitudes and hence the increase in SSWs."

· There are 27 sudden stratospheric warming events counted in the third winter, but there are only 20 ensemble members, so there is more than one SSW event in one member. What is the definition of the SSW event used here? Is it only based on the U50? Is it reasonable to count like this? More clarifications are needed. A figure showing the wind changes and marked SSW events in each member would make it clearer.

We define the SSWs according to Charlton and Polvani (2007) using 10hPa zonal winds. This has been detailed more clearly on Pg 8 (L216-220) according to the following:
"We identify SSW events by using an algorithm following the procedures described in Charlton and Polvani (2007), where mid-winter sudden warming events are determined to take place if the 10 hPa zonal-mean zonal wind at 60°N becomes easterly. Once a warming is identified, no day within 20 days of a central date, defined as the first day in which the daily mean zonal mean wind at 60N and 10hPa is easterly, can be defined as an SSW."

We have also added figures showing the number of SSWs in each ensemble member for all three winters. In addition we have added a figure showing the zonal mean zonal winds at 10hPa for both cpl and atm-only where the zonal wind weakening is more clear in the second winter and also in the end of year 3.

· Discussions needs to be largely improved. Comparison with other related studies and limitations of this study are limited.

The Discussion chapter has been through substantial changes, with parts removed and added to the Results section while adding comparison studies to put our results more into context of the literature. However, we want to emphasize that other studies regarding

volcanic eruptions and SSWs are not to be found in the literature and thus we cannot use direct comparisons in those cases.

· The structure and English writing of the manuscript can be improved. The discussion and summary can be separated with improved discussions and clear conclusions in the summary.

We have made some logical changes to the structure of the manuscript as well as the text in general and we are certain that it reads more easily than before.

**Specific comments**
L37-46: Too long for an introduction on this well-known statement.
This has now been cut short.

L51: "the aerosols tend to stay longer in the polar stratosphere (Graf et al., 1994, 2007)"
Correct? This study ("Initial atmospheric conditions control transport of volcanic volatiles, forcing and impacts" https://acp.copernicus.org/articles/24/6233/2024/) shows longer lifetime of volcanic aerosol in the NHET after tropical eruptions compared to extratropical eruptions.
Here we referred to the aerosols that are able to reach the polar stratosphere but since this was not relevant for this study, this has been removed.

L57-61: Are Tropical or NH extratropical eruptions described here? Any difference will it have after tropical and NH extratropical eruptions?
The introduction gives a short research background describing the different impact low-latitude and high-latitude eruptions can have on the atmospheric circulation, originating in the different spatio-temporal characteristics of the radiative properties of the volcanic aerosols. This has now been clarified further where we no longer mention NH extratropical eruptions but only NH eruptions.

L111-112: "that is comparable to the interactive chemistry model version"
Not convincing with this simple statement.
This has been addressed with the following sentence in lines L125-128:
"...which is computationally less expensive to run, but simulates dynamical stratosphere-troposphere coupling and stratospheric variability with skills comparable to the interactive chemistry model version (Smith et al. 2014)"

What is the aerosol module used in this model? How is the model's ability on simulating volcanic aerosol evolutions and NH high latitude dynamics like polar vortex, SSW etc.? These are important aspects that needs to be evaluated for this study.
The following line has been added (L128-131): "CESM1/WACCM4 uses the Community Atmospheric Model Radiative Transfer (CAMRT) to parameterize the radiative forcing where it has been shown to accurately represent stratospheric aerosols by f. ex. simulating the temperature response following Mt. Pinatubo in 1991 (Neely et al., 2016)."

We also added description on the stratospheric dynamics in line 127: "This has been addressed by adding to line X the underlined text: "…and stratospheric variability like SSWs and the polar vortex"

L126-127: Better moving this sentence to the previous paragraph where describing the coupled ocean and related runs.
Done.

L132: EVA is not height dependent in Toohey 2016.
The EVA output is available as an input for various models and on different model levels, where levels usually refers to pressure levels that can be interpolated to height. We keep the statement as it is based on an original description by Toohey et al. (2016).

L137-138: a midlatitude location at which latitude?
45° N. This has been added in line 150.

Figure 1 and L136-142: Not clear how this scaled is performed based on extinction and aerosol mass from the figure and the text description. What is the original EVA forcing?
We have now changed the Figure caption to make it more descriptive. The caption now reads: "Figure 1: Time series of the original EVA aerosol extinction output (1/km, black curve) and the aerosol mass of the volcanic forcing file of Neely et al. (2016) (kg/kg, blue dashed curve) used for deriving the linear scaling coefficient for the conversion of EVA output into WACCM4 input (kg/kg, red curve). The time series are normalized (mean=0, standard deviation=1) to allow comparison of time series with different units."

Figure 1 and L147-148: The months in Figure 1 and decline on Oct 1 is confusing, if the decreasing starts on Oct. 1 as written in the text, then Nov., Dec., Jan next year is used for the first winter calculation?
Correct, although the aerosols start to decline on oct 1st they are still fairly concentrated in the first winter.

L154: It can be quite different for a 45° N injection compared to a 65° N injection, how this assumption affect your results needs to be discussed.
We agree, it is important to discuss this where we have added the following to the Discussions (L604-619):
"As mentioned in the methods, we assume a similar lifetime of volcanic aerosols at 65° N as at 45° N. When considering the e-folding time in Toohey et al. (2019), a substantial aerosol decrease of about 43% occurs at 17km (a.b.s.) for an eruption at 60° compared to at 0°. However, since our experiment assumes a constant stratospheric injection over 5 months with the aim to simulate a long-lasting HL eruption compared to a single injection at low latitudes, the difference in the e-folding time between low and high-latitudes would be expected to decrease. Using CESM2-WACCM6 with interactive chemistry Zhuo et al., (2023) identified that although an eruption at 64° N did have a shorter aerosol lifetime

compared to at 15° N, they lead to stronger volcanic forcing over the NH extratropics although the resulting climate impacts did not last as long. In addition, one of their conclusions was that different duration and intensity of both tropical and NH extratropical eruptions can lead to different results, stressing that our 6 month long sulfate injection is not directly comparable with volcanic eruptions of shorter duration. Although the aerosol lifetime in our experiment might be exaggerated into the third year, our results do indicate that the polar vortex weakening in winter 2 appears to act as a trigger for further weakening that eventually leads to SSWs in winter 3. In order to confidently confirm such a delayed link, additional sensitivity simulations are required and thus we leave that for future studies. "

L162-167 Better to start description of cpl and then atm-only to keep it consistent across the whole paper.
We agree, where the text has been edited accordingly.

Figure 2 and L194-206: Better to adjust the order of fig 2(a) and 2(b) and related descriptions, same consistency reason. The time axes Jan, May, Sep. is confusing, better to use Jan. Jul. instead? Any significance test results?
The axis has been changed accordingly where their caption is now Fig. 2c) and d) and the colored area indicates 95% significance.

L196-199: Needs to be rephrased. The seasonal variation of the solar radiation is not the reason of different anomalies in the first and second winter.
Indeed it is correct, it now writes (L236-239):
"The perturbation of SW fluxes for both *cpl* and *atm-only* is influenced by the obvious strong seasonal evolution in solar insolation, where we see strong anomalies during the first summer north of 30° N than then becomes more confined to the mid latitudes as winter progresses with a slow decrease towards the end of the third year (Fig. 2c-d)."

Figure 2c: why is there a break in the tropics (around 0 degree) in the aerosol mass distribution?
This occurs due to the scaling with Neely et al. datasets. To clarify we have added the following to subchapter 3.1. (L245-250)
"Overall, the radiative forcing thus remains largely confined to NH extratropical summers with the exception of a slight significant increase around 30-60° S in the second and third summer (Fig. 2c-d) that is visible at around 14-15 km a.s. (Fig. 2b). This occurs due to spatial features in the Neely et al. (2016) aerosol forcing that we use for scaling, where a slight aerosol increase occurs at lower latitudes, although this is not detectable when the aerosol mass is averaged through the atmospheric column with respect to time (Fig. 2a)."

L210-211: "at 65°N" is confusing.
We agree, this has now been removed.

L212-213: Better to convert unit to avoid this exponential value expression.

Done

L218-223: Unclear explanation. More analysis are needed to differentiate the shortwave and longwave radiation effect and the direct radiation effect and dynamical effect to understand the different stratospheric temperature responses in the summer and winter.

We are certain that this has been addressed by adding the LW figure in addition to our T50 figures.

Figure 3 and L233-234: Is this the 2 STD of experiment or control ensemble runs? How different are they compared to the control runs? Figure 3a and 3b show different length, better to use Jan and July?
This would be the STD of the U50 anomalies (perturbed minus unperturbed) - significance has been added to figure (orange markers).

L237-240: "high latitude into midlatitudes" and "subtropics into midlatitudes", one is equatorward, oppositely, the other one is poleward, confusing.
This has been changes accordingly and now writes (L297-298):
"Similarly, the zonal wind weakens into the midlatitudes over the Pacific while it is stronger in mid to high latitudes over the Atlantic."

Figure 4: better to make it larger, it's not easy to see the details.
We hope it is possible to manage this during publication, if accepted.

L246-248: where shows the local heating? Figure 3a and 4a only shows the temperature difference to the control run, but what is the temperature gradients of experiment and control runs?
Our newly added longwave flux figure in addition to Fig. 3a shows the local heating from the volcanic aerosols within the stratosphere where the difference between pole and mid latitudes is clear for both variables, especially when considering the summer season where it continues into winter (Nov). This is evident of the role of thermal wind balance in the initiation of the polar vortex strengthening identified.

L255-259: "locate sources of wave activity"? But the cooling and the upward wave activity locates at different areas, then how can this explain the bottom-up mechanism?
The 2m temperature gradient show evidences of this, where a rapid increase and/or a retreat of the spatial T2m cooling emerges in the T2m gradient and is located in areas of increased Plumb wave activity (see subchapter 3.3.2 and figures therein, Pg 19-24).

L275-277: what is the direct thermal forcing?
This has now been removed.

L277-279: Do you mean the inconsistent results are due to the U50 definition, then why use this index? what if other indices are used, can they show consistent results?

We have indeed changed this and now we refer to the U10 (while adding U50 to the supplementary)  that shows the zonal wind weakening more robustly (see Fig. 3c-d).

L280-282: North Pacific ... North Atlantic and Siberia, they are all ocean, how to understand the reasoning "pointing to a possible influence of the change in land-sea thermal contrast"?
We have now edited this part and moved it under section 3.3 that we hope clarified this comment.

Figure 5: as written in the general comments, clarification on the definition/counting/presenting of SSW events are needed.
We agree and have added extra panels to (now) Fig. 6 that show the number of SSWs for individual ensemble member as well as including more statistical analysis to this part (Pg 15-17, L378-388).

L329-330: why is it a stratospheric cooling? If thermal response to aerosol injection, then it's stratospheric warming.
The thermal wind balance strengthens the stratospheric polar vortex and thereby the polar vortex encloses cold polar air.  We also see weak evidence that eddy feedback acts to sustain this strengthening (see section 3.3.1, Pg 17-19). The thermal warming is mostly dominant during summer but not winter and thus it can be considered a triggering factor in the polar vortex strengthening during winter. This has been explained better in L390-406: "According to the above, the evolution in *cpl* from winter 1 to winter 3 can be summarized as follows: In the first winter, the thermal forcing appears to be stronger than the upward wave flux because of the large amount of aerosols present, thereby dominating the response that causes the polar vortex strengthening and the inclusion of cold polar air within. In the second winter, the thermal forcing from the volcanic aerosols at midlatitudes has decreased where it is now mostly confined to higher latitudes as seen both in the LW flux and T50 (Fig. 2f and Fig. 3b). We suspect that in addition to the aerosol decrease, this slight decrease in the temperature difference between high and midlatitudes allows the strong upward wave flux to dominate and enter the upper stratosphere. There the waves are absorbed that causes further warming over the polar cap in addition to weakening the zonal stratospheric winds (Fig. 5b and Fig. 4b). This upward wave flux and weaker winds continue into the third winter, where winter 2 likely acts as a precursor, allowing for SSWs to develop more frequently as detected in the T50 warming that is now confined over the polar cap (Fig. 4c and Fig. 5c, respectively). The expected absence of a surface response is obvious in our *atm-only* experiment where basic physical mechanism, via the thermal wind balance due to radiative heating, dominates the atmospheric circulation response in the first two post-eruptive winters, with a strong stratospheric polar vortex isolating the cold air over the polar regions in the second winter as in the *cpl* experiment (Fig. 5a-b). "

Section 3.3: Does it contribute a lot to the main purpose of the paper by just describing the detailed spatial patterns. What connections do they have with previous results? How aerosol distribution lead to the temperature responses? How do they relate to the different

cpl and atm-only configurations? Addressing these questions are helpful to improve the quality of the study.

This section has been greatly improved with additional analysis and figures where we have tied these results better with the results in the stratosphere. We also moved the 850hPa Plumb flux from the Stratospheric section and into the Troposphere section (see Pg 17-24).

Summarizing discussions: The first three paragraphs just repeat most of descriptions in the results section. Discussions are needed to compare with other related studies. Like how's SSW response to volcanic eruption in the observations and studies using other models? What different reasons do they have if showing different results? How the specified chemistry model configuration affects the result? What kind of impact will it have on the results if including aerosol microphysics and stratospheric chemistry in the model? This study ("Volcanic forcing of high-latitude Northern Hemisphere eruptions" https://www.nature.com/articles/s41612-023-00539-4#:~:text=High%2Dlatitude%20explosive%20volcanic%20eruptions,Pinatubo%20eruption . ) shows initial polar vortex stability affects the aerosol distribution, but the forcing is produced with EVA, how will these affect the results. These needs to be discussed.

The Summarizing discussion chapter has gone through substantial changes where these factors mentioned have been addressed. Regarding the SSW response to volcanic eruptions in observations and other models, such studies have not been found in the literature by the authors although we do encourage such studies. We believe the rest of the RW comments here above are addressed in the following paragraph (L604-629):

"As mentioned in the methods, we assume a similar lifetime of volcanic aerosols at 65° N as at 45° N. When considering the e-folding time in Toohey et al. (2019), a substantial aerosol decrease of about 43% occurs at 17km (a.b.s.) for an eruption at 60° compared to at 0°. However, since our experiment assumes a constant stratospheric injection over 5 months with the aim to simulate a long-lasting HL eruption compared to a single injection at low latitudes, the difference in the e-folding time between low and high-latitudes would be expected to decrease. Using CESM2-WACCM6 with interactive chemistry Zhuo et al., (2023) identified that although an eruption at 64° N did have a shorter aerosol lifetime compared to at 15° N, they lead to stronger volcanic forcing over the NH extratropics. In addition, one of their conclusions was that different duration and intensity of both tropical and NH extratropical eruptions can lead to different results, stressing that our 6 month long sulfate injection is not directly comparable with volcanic eruptions of shorter duration. Unlike our simulated eruption using a version of WACCM4 where the chemistry is prescribed, natural volcanic eruptions can contain various chemical compounds that impact the formation and the lifetime of sulfate aerosols as well as affecting the atmospheric circulation via e.g. ozone depletion like halogens are known to do. More advanced versions as well as models that include interactive chemistry are thus important to reveal in more detail the chemistry-climate interactions that occur in the natural world (Clyne et al., 2021; Case et al., 2023; Fuglestvedt et al., 2024). Thus our idealized experiment can be considered primitive in the sense that it only considers sulfate aerosols

but sufficient when focusing on answering questions on the basic mechanism that such eruptions can initiate. "

L416-422: How this relates to equatorial eruptions? "WACCM4 is insensitive to the injection latitude", is the volcanic forcing produced by EVA? How can this conclusion be made here?
This has been removed

L427: what is "dynamic surface response"?
This has been removed

L430-432: Is this too arbitrary? No model-observation comparisons were made, and the aerosol forcing is much stronger than any volcanic forcing used in previous CMIP5/CMIP6 simulations.
Since this is mentioned as a possible caveat we do not think that we need model-observation comparison to make such a statement on the CMIP5 models in general (that includes CESM1). We also want to underline that stronger volcanic forcings have been used in previous simulations although most of them are injected at low latitudes (e.g. Zambri et al., 2019; Zhuo et al., 2024).

L434-435: Don't understand this conclusion. Figure 3 shows stronger stratospheric warming in cpl than atm-only in the second and third winter.
We believe that the stratospheric (thermal) warming identified in winter 2 acts as a trigger for the SSWs in winter 3 that does not occur due to volcanic thermal forcing. This has been highlighted in both the Discussions (L567-573) and Conclusions (Pg 27-28).

L439: what is "an intrinsic reason originating in the model"?
Our understanding is that this originates in the model parameterization of various physical processes.

 **Technical corrections**
L43-44: surface and stratospheric meridional temperature gradients?
done
L62: effect -> affect; as the positive phase of ... -> Leading to a positive phase of...?
Not relevant now
L81: Icelandic volcanism?
Volcanic activity in Iceland
L90-91: history and current activity makes these types of eruption?
We believe that the eruption history in addition to current activity in Iceland do make these type of eruptions an ideal reference case.
L96: the response within of NH stratospheric polar vortex?
Done
L194: short-wave -> shortwave
Done

L199: substantially

Not relevant now

L272: N America -> North America

Done

L462: smaller size -> smaller magnitude?

Done

Reviewer #2

We want to thank reviewer #2 (RW2) for their detailed and constructive comments where we are certain that it has led to important improvements of this study. Additional analysis has been done as well as making changes to the structure of the paper just to name a few. Here below we have addressed all of the comments made by RW2 where our responses are detailed in red. Since we have made considerable changes within the manuscript, part of the more detailed comments made by RW2 might no longer be relevant but in those cases we will refer to the changes made when appropriate.

This study assesses how high-latitude eruptions affect stratospheric circulation and more specifically the occurrence of sudden stratospheric warming (SSW) events. There are few studies on the influence of high-latitude eruptions on stratospheric circulation, which makes this a welcome contribution to the field. However, there are issues with the methodology that make me skeptical of the conclusion of an extremely significant impact in the third post-eruption year. I'm requesting this be better explained, kept from being the only focus of the title, and that a small amount of additional simulation output be presented to give a clearer sense of this conclusion's veracity. There are also issues with the structure of the paper and a number of instances where the results are insufficiently explained, which I would like to see the authors rectify, and so I am requesting major revisions. There are a number of positive aspects of this study, particularly the combined use of atmosphere-only and coupled simulations and the in depth presentation of wave activity anomalies, so I hope the authors will modify this manuscript to realize the potential of these inclusions.

Thanks, in the revision of the manuscript we have especially focused on more convincingly illustrate the impacts on the third post-eruption year and make the overall paper better structured and complete.

Major comments
1) I'm skeptical of the author's claim of a very substantial increase in SSWs specifically in the *third* post-eruption winter. The authors do present this result as "surprising" but since readers will assume what they will I want to be careful this isn't a methodological artifact. I hence request the authors give a clear explanation of why the effect is substantial only in the 3rd year and show a time series of the stratospheric wind from which it can be seen how distributed the SSWs are among ensemble members (in order to see how susceptible this result is to noise). There are a few issues here so I'd like to see more material:

   a) It is hard to believe this would be so strong *after* most aerosols have left the stratosphere, so this should be thoroughly explained, along with potential caveats.
The response in the third winter is stronger than expected, but our results indicate that the weakening of the polar vortex in winter 2 due to the aerosol decrease and the slight thermal warming present at high latitudes (Fig. 2f and 3b) act to trigger the SSWs in winter 3 (see L390-406). Indeed this could occur due to changes in wave propagation due to the aerosol

decrease that lead to wave breaking and SSWs at high latitudes. A similar but opposite response was detected in Bittner et al. 2016 (that the reviewer correctly suggested as a reference case) following a Tambora-like eruption. This we have added to the Discussions, see L584-595.

b) The large difference in SSW number across seasons in the control experiments suggests the 20 ensemble members used are not enough to make firm conclusions on post-eruption SSWs, which is potentially a major issue here.
We agree that the signal to noise is potentially lower than is needed to get a robust response where larger ensembles would be needed to confirm. The following sentence has been added to subchapter 3.2.3. (386-388):
"However, when considering the different number of SSWs between winters in the unperturbed experiment we cannot rule out the possibility that large ensembles are required to confirm this link."

c) As explained more below, the volcanic influence may be substantially weaker in the third year than the prescribed forcing used here represents.
This is a valid point and we have addressed the lifetime of aerosols at higher latitudes compared to at mid-low latitudes in the Discussions in the following paragraph (Pg 26, L604-619):
"As mentioned in the methods, we assume a similar lifetime of volcanic aerosols at 65° N as at 45° N. When considering the e-folding time in Toohey et al. (2019), a substantial aerosol decrease of about 43% occurs at 17km (a.b.s.) for an eruption at 60° compared to at 0°. However, since our experiment assumes a constant stratospheric injection over 5 months with the aim to simulate a long-lasting HL eruption compared to a single injection at low latitudes, the difference in the e-folding time between low and high-latitudes would be expected to decrease. Using CESM2-WACCM6 with interactive chemistry Zhuo et al., (2023) identified that although an eruption at 64° N did have a shorter aerosol lifetime compared to at 15° N, they lead to stronger volcanic forcing over the NH extratropics. In addition, one of their conclusions was that different duration and intensity of both tropical and NH extratropical eruptions can lead to different results, stressing that our 6 month long sulfate injection is not directly comparable with volcanic eruptions of shorter duration. Although the aerosol lifetime in our experiment might be exaggerated into the third year, our results do indicate that the polar vortex weakening in winter 2 appears to act as a trigger for further weakening that eventually leads to SSWs in winter 3. In order to confidently confirm such a delayed link, additional sensitivity simulations are required and thus we leave that for future studies."

d) The number of SSWs in year 3 cpl are *more* than 1-per-year on average, which seems strange. Could the authors please add a time series of the stratospheric wind in each ensemble member either to the manuscript or supplement, showing where these transitions occur? I want to at least know this isn't heavily affected by a handful of simulations that cross the SSW threshold several times, which would mean potentially far more than 20 simulations are needed. Likewise, perhaps another statistic (e.g. "percent of

ensemble members containing at least one SSW" in each winter) would be less susceptible to noise, so I would encourage the authors to attempt this if it helps establish the claim.

We have now added to Figure 6, where we show the number of SSWs in both perturbed and unperturbed experiments for winters 1-3 where, using a Kolmogorov-Smirnow test, the difference becomes significant at the 95% interval in winter 3. We do get more than one SSWs in 50% of the ensemble members in winter 3, underlining the strong response emerging in the *cpl* experiment. Two SSWs per year is indeed uncommon, although possible, and three almost unheard of except in models (Ineson et al., 2023). The following sentence has now been added to subchapter 3.2.3. (L378-388):

"When comparing the ensemble sum of SSWs in the perturbed and unperturbed experiment using a Kolmogorov-Smirnov test (Fig. 5d), a significant increase in the number of SSWs occurs in winter 3 (p=0.0135). This underlines the generally strong SSW response occurring in winter 3, when the fraction of ensemble members having more than 1 SSW per winter increases to 50% (10 ensemble members) in winter 3 compared to only 10% in winters 1-2. Of these 10 ensemble members, two members show three SSWs per winter that can be considered highly unlikely based on historical records:  Despite winters with more than 1 SSWs are considered unusual, examples do exist in the observational record of multiple SSWs in one winter, like the winter of 2009/2010  (Ineson et al., 2023). However, when considering the different number of SSWs between winters in the unperturbed experiment we cannot rule out the possibility that large ensembles are needed to confirm this link."

2) I feel the structure of the paper is currently inhibiting its potential, and that this would be a much nicer paper to read if the structure were changed. Currently the Results section is very dry, simply stating what is in the figure, year-by-year from one experiment to the other. Nearly all explanations of the results are instead in the Summarizing Discussions section. I strongly recommend merging much of the explanations into the Results, so that the reader immediately knows why the Results are important. The discussion section could then become less technical and more focused on the big picture concepts of high-latitude eruptions, resulting climate damages, predictability, etc, for which the results have relevance. And about the Results again, I quite like the combined use of atmosphere-only and coupled simulations but find the structure of the Results limits their effective use. I feel this would be better if instead of the atmosphere-only experiment having its own Results section after most results have been described, the atm-only and cpl experiments were described together, perhaps with one section on the surface cooling pathway and another on the stratospheric warming pathway. Currently there is no clear sense in Results how the atm-only experiments relate to the cpl (more realistic) case.

This has now been addressed within the manuscript according to this comment where the cpl and atm-only results are discussed together. We thank the reviewer for this important point and agree that this reads much better now. The Results chapter is now organized as follows:

**3. Results**
**3.1 Volcanic Radiative forcing**

**3.2 Stratospheric Response**
3.2.1 First post-eruption winter
3.2.2 Second post-eruption winter
3.2.3 Third post-eruption winter
**3.3. Tropospheric response**
3.3.1 The role of eddy feedback
3.3.2 The role of surface cooling
We have moved the explanations that were in the Discussions into the Results section and where the Discussions are now dedicated to summarizing our results and putting them into context with other studies within the literature in addition to discussing potential caveats that might impact our results. We also added a short Conclusion section to reveal the highlight points of this study and their relevance for future studies.

3) The prescribed aerosol forcing doesn't account for high-latitude eruptions leading to aerosols in the stratosphere for less time than tropical eruptions, due to entering the stratosphere far closer to descending stratospheric circulation. This is not necessarily huge but could nullify much of the 3rd winter effect by positioning this stage at the 2nd winter. Fig. 2g of Toohey et al (2019) shows with interactive aerosol modeling that eruptions at 56N have a 12% to 44% lower e-folding lifetime than similar eruptions in the tropics, depending on eruption season. I don't believe this lifetime difference is covered in EVA, and is stated to not have been factored into the conversion from 45N EVA data to a 65N eruption. I request the authors check how their volcanic forcing compares to high-latitude eruptions with interactive aerosol studies and at least explain this as a caveat.
*Toohey, M., Krüger, K., Schmidt, H., Timmreck, C., Sigl, M., Stoffel, M., & Wilson, R. (2019). Disproportionately strong climate forcing from extratropical explosive volcanic eruptions. Nature Geoscience, 12(2), 100-107.*
This is indeed important where this caveat has now been added to the Discussion section as well as underlining that our experiment is designed around a longer-lasting aerosol injection, unlike the experiments in Toohey et al. (2019), with maximum values spanning in total 5 months that could potentially counteract part of this decrease. We refer to our more detailed response to comment No. 1c since it is related.

4) Post-eruption Eliassen-Palm and Plumb flux anomalies are shown and described, and I think their inclusion is one of the main things that makes this study original for a high-latitude eruption case. However, these are barely put into the context of a) the volcanic forcing that causes the anomalies or 2) the net impacts of wave-eddy interactions on stratospheric circulation. I feel these results need to be physically explained within the Results and put into a context someone in the volcano-climate community without a geophysical fluid dynamics background can appreciate. Perhaps the authors can deduce how the studied volcanic circulation impacts are modulated by eddies/waves from the already cited Bittner et al (2016) as well as DallaSanta et al (2019), then considering how those results might vary for a high-latitude case. The Bittner study includes EP fluxes that look especially appropriate for a comparison to the present study's results.

*DallaSanta, K., Gerber, E. P., & Toohey, M. (2019). The circulation response to volcanic eruptions: The key roles of stratospheric warming and eddy interactions. Journal of Climate, 32(4), 1101-1120.*

This is an important point and has now been thoroughly addressed where we have added eddy feedback calculations (See Fig. 6 and Supplementary Fig. S4 in subchapter 3.3.1 and text therein) to support our findings as well as linking the EP and Plumb flux better to both our stratospheric and tropospheric results. Furthermore, a comparison to both Bittner et al. 2016b (Discussions) and DallaSanta et al. 2019 (subchapter 3.3.1) has been added according to the following paragraphs

L584-595: "Bittner et al. (2016b) identified a similar but opposite response, when compared to our *cpl* response in winters 2-3 (Fig. 4), following a Tambora-like eruption where a strengthening of the polar vortex due to less wave breaking at high latitudes was considered to be an indirect effect associated with a changes in planetary wave propagation. Since the volcanic aerosols in our experiments have declined extensively in the third winter, making the aerosol thermal forcing a limited factor, we cannot rule out similar indirect effects where changes in wave propagation leads to an increase in wave breaking at high latitudes and hence the increase in SSWs.

While not directly comparable to our study but still providing an important analog, Muthers et al. (2016) identified an average increase in the number of SSWs during a 30-year (constant) decrease in solar radiation in line with our significant increase in SSWs in winter 3. Our results do support the findings of Sjolte et al. (2019), where the stratospheric temperature gradient does not appear to play a major role in the polar vortex weakening we identify but rather the upward wave flux."

L441-448: "DallaSanta et al. (2019) reported on the role of eddy feedback in the polar vortex strengthening following a Pinatubo-like volcanic eruption. One of their conclusions were that the thermal-wind balance is too simplified in explaining the simulated stratospheric polar vortex response where eddies are needed to mediate atmospheric perturbation, both to couple the stratosphere to the troposphere as well as achieving the forced stratospheric response alone. Although our eddy feedback results suggest a similar response as in DallaSanta et al. the signal appears to be quite low compared to the noise as is depicted by the variability between winters in the unperturbed experiment."

5) I do not find the title suitable for this study, so hope the authors will alter it. There are three things about the current title I find problematic:

   a) Only a minority of the paper is really about SSWs and this is the most uncertain part of the results. I feel the paper doesn't concretely enough settle the SSW question to warrant this as a title. But the study would be more defensible if it generalized this part of the title to impacts on "stratospheric circulation and sudden stratospheric warmings" or just "stratospheric circulation", or similar, maybe a Part 1 style title given the mention of an upcoming study also on high-latitude eruptions and circulation.

   b) I would take out "long-lasting", as it's just confusing in that it sounds like this is a

constant-aerosol-presence geoengineering experiment. I don't believe distributing the explosive eruptions over 6 months is a major factor for the results, compared to a one-off eruption of the same mass, and there isn't a one-off experiment here to compare with anyhow. I would omit this and let readers simply read the methods for an explanation that this is designed to closely resemble eruptions like Laki.

We understand the reviewer's point, although we do want to keep the long-lasting part at this stage. When concerning the changes that have been made we still feel this part is relevant.

 c) I also feel "an idealized modeling study" is confusing. I get that this is not an actual eruption case, but this level of being idealized is not particularly high. This subtitle makes it sound like the study uses an energy balance or intermediate-complexity model, rather than a full GCM. I would omit or reword, as the idealized nature is explained in the abstract anyhow.

We agree, in addition to changing our manuscript title that now reads:

"Stratospheric circulation response to long-lasting Northern high-latitude volcanic eruptions"

Furthermore, we have removed the focus on both the "idealized" and "long-lasting" part and added the following sentence to the Methods section to clarify (L162-170):

"In this experiment we assume stratospheric injection only, although similar eruptions in the natural world would likely inject part of the total aerosol mass within the troposphere during the eruption. Past NH eruptions like Laki and Eldgjá had an atmospheric loading of 210Tg and 120Tg respectively, much larger than our 14Tg eruption, that was carried aloft with the eruptive column up into the upper troposphere with portions of the aerosols reaching the lower stratosphere during the eruptions (Thordarson et al., 2001). Hence our experiment can also be considered as a 6-month stratospheric aerosol injection that is analogous to similar although smaller eruptions (as compared to Laki) without the tropospheric aerosols."

Specific comments:
Line 29: For multiple reasons it seems doubtful the studied impacts are an "important source of interannual variability and a possible source of increased seasonal predictability of northern hemisphere regional climates": 1) eruptions of this type and magnitude only occur 2-3 times per millennium, 2) adding similar magnitude eruptions to forecast systems has small influence on prediction skill (Aquila et al, 2021), and 3) as this study mentions, simulated volcanic impacts tend to be overestimates, and still the results aren't incredibly confident. I expect there are more realistic reasons to study this, e.g. to understand impacts on people and ecosystems on the rare occasions when these events do happen (both historical, e.g. after Laki, and future).
*Aquila, V., Baldwin, C., Mukherjee, N., Hackert, E., Li, F., Marshak, J., … & Pawson, S. (2021). Impacts of the eruption of Mount Pinatubo on surface temperatures and precipitation forecasts with the NASA GEOS subseasonal-to-seasonal system. Journal of Geophysical Research: Atmospheres, 126(16), e2021JD034830.*

We see the reviewer's point and agree but we do want to emphasize that more sensitivity studies are needed to evaluate the link between volcanism and SSWs, both regarding size and latitude. If a link is established between volcanism and SSWs, that in itself could at least play a part in increasing predictability of what to expect following the eruption. We have removed "interannual variability" and focus on the need to study different eruption magnitudes according to the following (L27-31):

"The identification of a deterministic response such as increased SSWs following high-latitude volcanic eruptions calls for increased attention given the widespread and prolonged surface cooling SSWs can initiate and thereby affect societies throughout the continental NH. In addition, the sensitivity of such events to the eruption magnitude needs to be evaluated in terms of a possible source of increased seasonal predictability of northern hemisphere regional climates."

Lines 51-2: The line "aerosols tend to stay longer in the polar stratosphere" appears incompatible with the aerosols entering the stratosphere far closer to the downwelling polar circulation, and the shorter lifetime found in studies using models with interactive aerosol microphysics and chemistry (e.g. Toohey et al., 2019 cited above). Please rectify or explain this.

Here we referred to the aerosols that are able to reach the polar stratosphere ( (Graf et al., 1994, 2007; Sun et al., 2024)) but since this was not relevant for this study, this has been removed.

Line 52: I'm not convinced the tropopause being lower near the poles would enhance the dynamical effects of aerosols there compared to lower-latitude eruptions, since the circulation structure tends to follow relative heights in the troposphere rather than actual geometric heights. Could the authors at least please explain this in their answer to the review?

Similar to to the above, we have removed this part since it is not directly relevant here.

Lines 54-5: "not analogs" is a bit extreme, as certainly tropical and high-latitude eruptions are related. How about "not close analogs"?

Agreed, this had been edited accordingly (L51).

Line 64: Somewhere in the intro the manuscript should make clear what's original in this study. The focus on high-latitude eruption impacts on SSWs seems to be new to the best of my knowledge, and same for the focus on eddies (EP-flux analysis) after a high-latitude eruption. Studies on related SSW impacts should be cited, for instance the impact of reduced sunlight studied by Muthers et al., 2016 is relevant here. Also, for high-latitude eruptions and stratospheric circulation (but not SSWs), The 'Part 1' Zambri et al (2019) and Oman et al (2005) are relevant. I'm also a bit surprised the results of Sjolte et al (2019) aren't discussed more specifically, as would seem appropriate.

*Muthers, S., Raible, C. C., Rozanov, E., & Stocker, T. F. (2016). Response of the AMOC to reduced solar radiation–the modulating role of atmospheric chemistry. Earth System Dynamics, 7(4), 877-892.*

*Oman, L., Robock, A., Stenchikov, G., Schmidt, G. A., & Ruedy, R. (2005). Climatic response to high‑latitude volcanic eruptions. Journal of Geophysical Research: Atmospheres, 110(D13).*

*Zambri, B., Robock, A., Mills, M. J., & Schmidt, A. (2019). Modeling the 1783–1784 Laki eruption in Iceland: 1. Aerosol evolution and global stratospheric circulation impacts. Journal of Geophysical Research: Atmospheres, 124(13), 6750-6769.*

We agree, these references are important where Oman and Zambri papers have now been cited in the text.

Muthers et al. is now discussed in the Discussions in relation to our results according to the following (L591-593):

"While not directly comparable to our study but still providing an important analog, Muthers et al. (2016) identified an average increase in the number of SSWs during a 30-year (constant) decrease in solar radiation in line with our significant increase in SSWs in winter 3."

We have also mentioned the much relevant results of Sjolte et al in the following sentence in the Introduction (L75-78):

"An example of this bottom-up mechanism following HL eruptions is demonstrated in Sjolte et al. (2021) where they linked a weak polar vortex to an increase in wave energy flux from the troposphere to the stratosphere without the meridional stratospheric temperature gradient playing a role."

As well as in the Discussions (L593-595):

"Our results do support the findings of Sjolte et al. (2019), where the stratospheric temperature gradient does not appear to play a major role in the polar vortex weakening we identify but rather the upward wave flux."

Lines 75-8: Could the authors please explain the bottom-up method as clearly in the text as they do the top-down method? Is this also a thermal wind, but in the opposite direction due to lower tropospheric cooling, compared to stratospheric warming? These "bottom-up" and "top-down" terms aren't used again, despite their relevance to the cpl and atm-only experiments that I think should be related to these terms (or at least the language made consistent).

We agree where we have edited the Introduction with the following paragraph to demonstrate examples of both mechanisms:

L63-73: "A strengthened polar vortex can affect the troposphere as the positive phase of the Northern Annular Mode), while a weaker polar vortex is linked to increased likelihood of sudden stratospheric warming events (SSWs) in the stratosphere and a negative Northern Annular Mode in the lower troposphere (Haynes, 2005; Domeisen et al., 2020; Huang et al., 2021; Kolstad et al., 2022, and references therein), demonstrating an example of a top-down mechanism.

With its origin in the noisy stratosphere, this top-down mechanism can result in tropospheric signatures following volcanic eruptions. However, the signature tends to be weak in both observations and numerical simulations due to the different realizations and advanced statistical methods needed to extract the signal from the noise (Weierbach et al., 2023; DallaSanta and Polvani, 2023; Kolstad et al., 2022; Azoulay et al., 2021; Polvani et al.,

2019; Zanchettin et al., 2022; Toohey et al., 2014). The radiative surface cooling following large volcanic eruptions has been shown to affect the stratospheric polar vortex via a bottom-up mechanism (e.g., Graf et al., 2014; Peings and Magnusdottir, 2015; Omrani et al., 2022). An example of this bottom-up mechanism following HL eruptions is demonstrated in Sjolte et al. (2021) where they linked a weak polar vortex to an increase in wave energy flux from the troposphere to the stratosphere without the meridional stratospheric temperature gradient playing a role."

We have also used these concepts when summarizing the main points in the conclusions.

Lines 85-9: Explosive vs effusive. Since this paper is about stratospheric aerosols it is about explosive eruptions. Laki was a relatively long-lasting eruption event in the 1780s, but what's important here is several explosive eruptions that emitted material into the stratosphere over a period of 5 months in the 1780s, not effusive emission into the troposphere. This should be explained if the long-lasting element is a focus, and if not should the discussion here should at least be clearer. The 'Part 2' Zambri et al (2019) study includes a table of these eruptions from an earlier source
*Zambri, B., Robock, A., Mills, M. J., & Schmidt, A. (2019). Modeling the 1783–1784 Laki eruption in Iceland: 2. Climate impacts. Journal of Geophysical Research: Atmospheres, 124(13), 6770-6790.*
This is a valid point and we have now added the following sentence in the Introduction (L99-102):
"During part of the eruption time such eruptions can become explosive (referred to as mixed-phase eruptions) when ascending magma in a conduit comes in contact with water as was considered the case with both Eldgjá and Laki, explaining their widespread impacts."
Here, we also refer to our answer to comment No. 5c.

Fig. 1: I cannot understand the "normalized" units. Can the authors please switch this to something that clearly indicates how far the state is from peak and zero response? Preferably this would simply be the actual units, and the right side of the plot can be used as a secondary axis, allowing the same figure to show both kg/kg and 1/km.
Since we are only interested in comparing these different profiles, we have included additional explanations on the normalized units used. We hope it offers more clarity than before. It now writes:
"Figure 1: The time series of the original EVA aerosol extinction output (1/km, black curve) and the aerosol mass of the volcanic forcing file of Neely et al. (2016) (kg/kg, blue dashed curve) used for deriving the linear scaling coefficient for the conversion of EVA output into WACCM4 input (kg/kg, red curve). The time series are normalized (mean=0, standard deviation=1) to allow comparison of time series with different units. The horizontal axis is time (months) from the start of the eruption. Here we assume that the aerosol lifetime at 65° N is the same as at 45° N. Dashed vertical lines show the three winters that we focus on in this study."

Line 153: Is the "red curve" actually the orange one?
Indeed it might appear to be orange in some monitors.

Fig. 2a,b: Can the x-axis please be edited to prominently show every January?
Yes, this has been changed accordingly.

Line 236: Since there's a lot going on here, could the authors please introduce this subsection with a brief explanation of how this will fit together, e.g. this is a time where stratospheric warming (rather than tropospheric cooling) is a prominent factor, and presumably is responsible for altered wave activity and through this circulation changes.
Now we have added a more thorough text in section 3.2. Since that section covers the stratospheric response, we begin by describing the newly added long-wave figure that depicts the long-wave absorption of sulfate aerosols that then warm the stratosphere that then is followed by subsections of the response initiated by the forced warming response. This now reads (Pg 10, L261-265):
"The strong seasonality in the LW perturbations described above also characterizes stratospheric temperatures, where a strong increase in the zonally averaged temperature at 50 hPa (T50) is detected north of 30° N during post-eruption summers in both experiments (Fig. 3a and 3b)."
We also added the following descriptive text to subchapter 3.2.1 (L304-307) to underline this factor.
"Therefore, the local heating due to the volcanic aerosols and the associated increase in the meridional temperature gradient in the stratosphere appear to dominate the eruption dynamics of the polar vortex via thermal wind response, also depicted by the LW anomalies (Fig. 2f)."
Similar work has been done for the Tropospheric section.

Line 243: Could the authors please explain physically the significance of "strong upward EP flux" (i.e. in terms of meridional buoyancy flux and wave-mean circulation interactions). Not many researchers in the volcano-climate community have extensive geophysical fluid dynamics training. Please generally walk the audience through.
We agree, the wording here was a bit abstract. We hope that both groups will now understand, where this now reads (L298-304):
"The strong upward EP flux (black arrows) is an indicator of the direction of propagated waves originating at the surface around the midlatitudes, where the horizontal and vertical EP flux components can be considered proportional to both eddy heat and momentum flux (Fig. 4d). A convergence (negative divergence, dashed red contours) in the wave flux is detected in the upper troposphere that acts towards weakening the tropospheric westerlies (Fig. 4d) while the EP flux and its convergence within the stratosphere does not appear to impact the stratospheric mean flow and the polar vortex. "

Line 255: Could the authors please explain what the significance is of the *near-surface* Plumb flux anomalies? Does this quantity link surface cooling to a stratospheric mean flow response that is modulated by eddies? Given forcing of the mean flow by waves/eddies is

the divergence of wave flux activity, how does this relate? This is perhaps tricky, because the temperature responses in a-c don't overlap well, while there's evidence that polar circulation responses to temperature changes occur non-linearly and differ strongly depending on the location of the forcing. But I hope the authors can offer a little perspective on this.

*De, B., Wu, Y., & Polvani, L. M. (2020). Non‑additivity of the midlatitude circulation response to regional Arctic temperature anomalies: The role of the stratosphere. Geophysical Research Letters, 47(16), e2020GL088057.*

This is a strong response indeed where our results do not show a clear source. Since it calls for further analysis on the surface response, we discuss this in section 3.3.2 where we speculate that it originates in the spatial T2m pattern identified in winter 1 according to the following sentence (Pg 22-23, L526-536):

"Both the zonal and meridional Tgrad components show an increase in the northern part of Alaska that coincides with the region of T2m warming over the Aleutian/Alaska region (Fig. 7a) and the strong upward Plumb flux (Fig. 4g). This warming in addition to the strong continental cooling over North East America and the general decrease in Tgrad spanning from mid to northern part of North America, might also trigger this strong Plumb flux in the Northern Pacific. At least it is unlikely that the Tgrad alone could explain such a strong increase in the upward Plumb flux where the source is likely to be rooted in anomalous spatial temperature patterns. Although not shown, an increase in sea ice extent in East Siberia extending into the Chukchi Sea, in addition to the above mentioned temperature dipole over Alaska and North East America, might further support the role of the anomalous spatial temperature pattern occurring in the vicinity of the strong upward Plumb flux detected."

We also see evidence of an increased eddy heat flux in the strong upward EP flux in addition to the Plumb flux increase in the North East Pacific. This is also supported in the increase in the zonal mean eddy feedback between 45-60° N for winter 1, implying that eddies are interacting with the mean flow that increases eddy generation that further acts to sustain the polar vortex strengthening. The convergence regions in the tropospheric mid latitudes indicate the slowdown of the tropospheric westerlies as is identified in the blue region at mid latitudes from the lower troposphere up into the stratosphere. The divergence region in the polar stratosphere is also evident of the detected zonal wind strengthening in addition to the increase in eddy feedback (Fig. 4c). This is indeed a bit tricky as the reviewer mentions so we hope this has clarified at least the main parts.

Line 259: Could the authors please add a line to explain what the significance is of the anomalies in wave activity source, and why it might be so focused on the North Pacific? This relates to the comment above, where the source might lie in the spatial T2m pattern and/or the Tgrad response, where the response is strong enough to persist all the way into the stratosphere.

Line 265: The caption should explain what the blue-to-red colored areas are. Presumably from the color bar units and match to the black contour lines this is a zonal wind anomaly. We thank the reviewer for noticing this. Now this reads:

"Figure 4: Winter stratospheric response in the *cpl* experiment. a-c) U50 (contours) and T50 (shading: red = warming, blue = cooling) response for winters 1-3, respectively. d-f) EP flux (arrows) and divergence (red contours) response, along with zonal-mean zonal wind response (black contours and shading: red = strengthening, blue = weakening) and climatology (green contours) in winters 1-3, respectively. g-i) Vertical component of the Plumb flux response at 850 hPa for winters 1-3, respectively. Contours and color-shaded areas indicate 95% significance according to a student's t-test. Only vectors that are significant at the 95% confidence interval are shown."

Line 271: Can this shift be attributed to the gradual development of surface cooling? Would be helpful if the authors can briefly rationalize this change between winters 1 & 2.
We suggest that the surface cooling does play a role but that the trigger lies in the slight decrease in the temperature difference between pola and mid latitudes as is both detected in the T50 (Fig. 3a) and LW flux (Fig. 2f). The upward wave flux then contributes to this zonal wind weakening and increase in T50. Since we have moved the 850hPa Plumb flux to the Troposphere this is no longer relevant here.
We also refer to the author's answer here below that should shed more light on this.

Line 280: As with the above, I'd like to see a brief physical explanation of how this information on the near-surface Plumb flux relates to the volcanic forcing and stratospheric circulation.
In general we have improved the text within the results section and now the following paragraph has been added to explain this link further (L500-504):
"In addition to this upward flux, we also detect downward propagating wave flux over both the Aleutian and Greenland regions at 850 hPa and over a large area south of 45° N at 150 hPa. This downward Plumb flux is evidence of changes in the planetary wave structure where wave reflection occurs due to the sudden weakening of the zonal winds identified in the U10 (Fig. 3a)."
We also added the following sentence in L343-347:
"Similar wave reflection pattern is known to be associated with SSWs, where we suspect that the decrease in the T50 difference between mid latitudes and the pole can act as a trigger for a weaker polar vortex in addition to the absorption of upward propagating waves into the stratosphere that is known to cause warming over the polar cap (Kretschmer et al., 2018). We will see further evidence of this in the next section."

Line 288: Somewhere in this paragraph should express that the aerosol has mostly left the stratosphere, while ocean cooling (possibly prolonged by interactions with sea ice) is still at play.
We agree where we have added the following sentence in the beginning of subsection 3.2.3 (L351-353):
"The SSW-like pattern of winter 2 clearly continues into winter 3, where most of the volcanic aerosols have decreased to the extent that their radiative impacts no longer dominate, except that the T50 warming is now confined over the polar stratosphere (Fig. 4c)."

Line 290: Please explain (or at least speculate on) why the wave activity flux is now purely upward unlike before.

We do thank the reviewer for pushing us further. This purely upward wave flux suggests that these waves are being absorbed within the stratosphere that causes warming over the polar cap, a pattern that behaves much like absorbing SSWs as defined by Kodera et al. (2016). Our SSW detection, although noisy, acts as further evidence of this response. See our changes in L343-347 as above.

Fig. 5: Considering the unperturbed case has no eruptions, shouldn't the first, second, and third winters all have similar numbers of SSWs? Here they are shown to vary from 6 to 15 (a factor of 2.5x), which suggests the 20 ensemble members are not sufficient for an SSW analysis, at least with the used methodology.

We agree, our SSW analysis along with the eddy feedback calculations do suggest that more ensemble members would be required to get a robust response. We have addressed this in section 3.2.3 by adding the following sentence (L386-388):

"However, when considering the different number of SSWs between winters in the unperturbed experiment we cannot rule out the possibility that large ensembles are needed to confirm this link."

We also mention this in the Discussions section according to the following (L574-582):

"Although the above coincides with a positive (negative) eddy feedback in the first (second) winter that could in theory play a role in sustaining the strengthening (weakening) of the polar vortex, our eddy feedback results indicate low signal to noise where further studies with additional ensemble members would be required to confirm their role in the forced response. Similarly as the eddy feedback, low signal to noise is also evident in the SSW analysis, both of which suggest the need for more ensemble members in order to get a more robust response. However, the response we do detect in the U50 and T50 fields is strong compared to the unperturbed run where the eddy feedback, and especially the SSW, provides an explanation in agreement with the patterns detected although noisy."

Line 302: Considering how noisy the data is I find this tiny p-value entirely misrepresentative.

Correct, we acknowledge this in the comment above..

Line 320: I agree with the statement, but I think it should be clearly stated atm-only has stratospheric warming but with minimal surface and lower-tropospheric cooling.

Indeed we agree, the following sentence is now in the end of section 3.2.1 that should clarify this (L307-311):

"Winter 1 in *atm-only* shows a similar thermal wind mechanism at play in the stratosphere as for the *cpl* experiment (Fig. 5a and 4a, respectively), in this case with the obvious less tropospheric influences - due to lack of forced surface cooling - as seen in the limited anomalous upward wave activity detected by the EP flux diagnostics (Fig. 5c). "

Lines 330: I would be a bit more specific on what the "stratospheric thermal response" is here and generally. I think the authors mean the aerosol warming here but heat fluxes are also part of the dynamical response.
We agree, the wording has changed in the edited version.
Line 333: Is there a specific reason there's no SSWs shown from the atm-only runs?
Since we did not detect a weakening in atm-only we decided it would be unnecessary.

Fig. 7: This figure interrupts the flow of the dynamical Results, so I feel it might be better suited in 2.3 Experimental design.
We agree that it does not below here and have added it to the supplementary (Supplementary Fig. S5).

Lines 364-76: I'm not seeing much evidence of cause and effect links from stratospheric changes to tropospheric responses in this section, which I note can occur due to SSWs or otherwise. I wonder if the authors can rework this a bit and perhaps relate it the literature on stratosphere-troposphere coupling.
In this section, we have added additional analysis where we compute the 2m temperature gradient (Fig. 8a). We also evaluate the average gradient for various areas with respect to the average number of SSWs in each winter (Fig. 8b). There we do see that the major changes occur in areas where the spatial cooling pattern either decreases (North East US) or increases (Barents Sea). This occurs where we also see an increase in the Plumb flux, suggesting where the wave sources are. Thus the newly added section 3.3.2 should shed more light on this, we e.g. say (L492-517):
"Apart from the magnitude of the cooling, the main difference between the surface temperature responses in *cpl* and *atm-only* is the presence of anomalous warming-cooling dipoles, hence regions of enhanced temperature contrast like the Aleutian/Alaska region. The vertical component of the 3D wave activity flux (the Plumb flux) at 850 hPa (Fig. 7d-f) allows us to locate the origins of the upward *cpl* EP flux (Fig. 4d-f) as being strongest over the north eastern part of the Pacific Ocean (off the west coast of North America) in winter 1, where it continues up into the stratosphere at 150 hPa (see Supplementary Fig. S1a). In winter 2, the vertical Plumb flux has decreases in the North Pacific and increases over the North Atlantic and Siberia, pointing to a possible influence of the change in land-sea temperature contrast (Fig. 7e). In addition to this upward flux, we also detect downward propagating wave flux over both the Aleutian and Greenland regions at 850 hPa and over a large area south of 45° N at 150 hPa. This downward Plumb flux is evidence of changes in the planetary wave structure where wave reflection occurs due to the sudden weakening of the zonal winds identified in the U10 (Fig. 3a).
An upward wave-activity flux now dominating both at 850 (seen in Fig. 7f) and 150 hPa (Fig. S1c), where it encircles the polar stratosphere north of 60° N.
Only minor activity is occurring in the Plumb flux of *atm-only* in winter 1 as expected, where downward flux dominates the mid-latitudes, with the exception of the upward flux over Greenland and the Himalayas that is most likely of orographic nature (Fig. 8c-d).

When the cooling is no longer confined to the NH mid-latitudes in *cpl*, and has reduced towards the polar regions as in winter 3, the upward Plumb flux also decreases compared to previous winters (Fig. 7f). This suggests that the mid-latitude spatiotemporal cooling pattern plays a part in the strong wave activity detected. This can be revealed by computing the T2m gradient (Tgrad) where strong land-sea temperature gradients are known for their ability to influence atmospheric wave activity (Hoskins and Valdes, 1990; Brayshaw et al., 2009; He et al., 2014; Wake et al., 2014; Portal et al., 2022)."

Lines 443-458: Optional, but since the manuscript does not really focus on the QBO I feel this could be better as a text in the Supplement with just a brief mention here.
We agree. This has now been shortened and discussed relative to the importance of initial conditions, now it reads (L629-640):
"Another important aspect that we do not focus on in our study is the role of different initial conditions on the forced climate response, where initial atmospheric and climate conditions, including e.g. the stability of the polar vortex, control the lifetime and distribution of the volcanic aerosols as well as the forced dynamic climate response (Weierbach et al., 2023; Zhuo et al., 2023; Fuglestvedt et al., 2024). An exception is our assessment on how the easterly and westerly phase of the Quasi-Biennial-Oscillation affect our results. We compared ensemble members showing easterly phase with the westerly ones to test if the U50 and T50 response patterns would be different. They were not, both phases showed a weakening of the U50 although the zonal winds were more confined and consistent over the higher latitudes of the NH during the easterly phase (not shown). The difference in the number of ensemble members used for these calculations could of course impact the statistics of this test of ours but not the overall pattern detected."

Lines 463: As I explained in the comment on Line 29, the probability of better predicting decadal variability through this study's explorations is quite low for multiple reasons. I recommend at least better contextualizing this prediction aspect. Alternatively and optionally, I would personally find the discussion/conclusion section more interesting if it instead focused on what the results suggest is experienced by people, societies, and ecosystems on the rare occasions when these eruptions do occur, e.g. understanding and remaining knowledge gaps for how stratospheric circulation responds to eruptions and through this affects northern latitude near-surface conditions.
We agree - and given the substantial changes within the manuscript this has now been removed.

Typographic/wording issues:
Line 38: language is awkward here, would change from "possibly very strong" to "at times strong" or similar.
This now reads :"The enhancement of the stratospheric aerosol layer, which typically occurs following strong sulfur-rich explosive volcanic eruptions, is an important driver of natural climate variability by imposing short-lived yet possibly very strong radiative

anomalies within the atmospheric column (Robock, 2000; Timmreck, 2012; Zanchettin, 2017)."

Lines, 41,42, and 194: "short-wave" and "long-wave" aren't usually hyphenated. Would cut the hyphens or at least make "longwave" in Line 217 consistent.

Done

Line 62: "affect" rather than "effect"

Done

Fig. 2b: the pressure coordinates are jammed together, so the figure should be altered to fix this.

Ok.

Line 458: Would be better with a paragraph break after "detected."

Since the structure of this part has changed this is no longer relevant

---

## Author Response (AR2)

Reviewer #2 - Report #1
We thank the reviewer for raising these important issues. We are certain that our response to both these issues and comments as well as the ones raised in report #2 has resulted in valuable improvements regarding the reliability of our results.

In addition to the reviewer comments and general changes associated with those comments, we made three changes to the manuscript that was not part of the reports. Firstly, we added an explanation behind the LW/SW difference between cpl and atm-only. That is mentioned in subchapter 3.1. (L368-371):
"We also detect a slight difference in the LW and SW fluxes that arises from differences in high cloud cover between *atm-only* and *cpl*, where *cpl* shows a decrease in high cloud cover in the northern high-latitudes, compared to *atm-only*"
and, in the Discussions (L676-681):
"Interestingly, a slight difference between *atm-only* and *cpl* is detected in both the LW and SW flux that is caused by a strong significant decrease in high cloud cover in the *cpl* simulation (not shown). This cloud cover decrease, especially at mid- to high latitudes, agrees with the increased LW fluxes at higher latitudes in addition to the decrease in SW flux and the associated surface cooling. This raises a question on the role of forced surface processes in these high cloud changes but since it is out of the current scope we leave it open for further studies."

Secondly, we also changed the scale in the Tgrad figure (Fig. 10) to improve clarity.

Finally, we added panels showing the tropospheric zonal wind anomalies at 200 hPa (U200) for cpl and atm-only to figures 8-9 in addition to a figure showing changes in waveguides (Figure 10) (shown as a probability of favorable propagation for Rossby waves). We did this since the nature of the strong tropospheric response detected and the equatorial movement of the subtropical jet should result in changes in wave activity. This we confirm, where we show that upward wave propagation is more favorable in winters 1-2 in addition to a waveguide decrease in the subtropical troposphere that also favors large-scale waves to be redirected towards the pole.
We added the following in the Tropospheric section (L527-555):
"Since upward wave activity depends on wave mean flow interactions, several factors are at play to explain the strong response in *cpl* vs *atm-only*. First, the change in zonal flow is substantially different between the two pairs of experiments, as shown by the U200 anomalies (Fig. 8g-i and Fig. 9e-f). In the first two winters we observe an intense deepening of the Aleutian low in *cpl* (Fig. 8a-b) associated with a large equatorward shift of the subtropical jet over the North Pacific (Fig. 8g-h, also seen in the zonal mean averages of Fig. 4). The change in zonal flow is not as large in *atm-only*, where there is a general decrease of U200 on the poleward side of the subtropical jets, rather than a marked equatorward shift

as in *cpl*. This further emphasizes that amplified surface coupling when the ocean is coupled to the atmosphere has a dramatic impact on the amplitude of the tropospheric response. Because the zonal flow acts as a waveguide for large-scale planetary waves, we expect changes in upward wave propagation in the stratosphere. To measure how waveguides change, Fig. 10 shows the probability of favorable propagation conditions for large-scale stationary waves number k=1,2,3, in function of pressure and latitude (see section 2 for the method). Areas of high probability show where large-scale waves preferentially propagate, while low probability regions indicate where wave propagation is prohibited. Generally, the mid-latitude troposphere is favorable for wave propagation, unlike the high-latitudes, consistent with the tendency for stationary waves to propagate upwards and to be deflected towards the equator, in climatology. After injection of the volcanic forcing, both cpl and atm-only exhibit an increase in the probability between 40 and 60°N in the lower stratosphere, but the responses in the troposphere are markedly different. In *atm-only*, wave propagation is inhibited in the free troposphere north of 60°N, for both winters 1 and 2 (Fig. 10d-e), which is consistent with the EP-flux anomalies of Fig. 5. This response is absent from *cpl* during winter 1 (Fig. 10a), and opposite during winter 2 when an increase of favorable conditions for wave propagation is observed (Fig. 10b). This increase in favorable conditions for wave propagation in the troposphere between 60 and 80°N persists during winter 3 in cpl (Fig. 10c), which is a partial explanation for enhanced upward wave propagation in the stratosphere described in Fig. 4f."

We also added to the Abstract, Methods, Discussions and Conclusions accordingly. Since this is much more significant to our study, we have shortened the eddy-feedback part and put the figure in SI.

Then the reviewer's comments (in black) and the associated response (in red) are listed here below:

The study is noticeably improved. There are a few smaller remaining issues I describe below, but if the editor finds that the authors sufficiently address these I would support publication without needing to see the manuscript again.
We thank the reviewer for these encouraging words.

Main comments

The study is now more appropriate in less strongly emphasizing the SSW impacts than before, as there isn't enough data to rigorously settle this here (i.e. too much noise, too few simulations). I pinpoint a few remaining lines below that I find don't sufficiently convey this context to the reader. I encourage the authors to be very upfront about the issues here, which could help guide future research on these dynamical effects.

Some of the key lines are worded insufficiently clearly or confusing. I point out a number of these in the abstract and a few in the text below. There are also a couple plot issues (two Fig. 6s, tiny font size in one of those).

Thank you, we have now changed the font size and the figure numbers.

Specific comments

line 16: "beyond the duration of the radiative forcing" is not representative of this study, as most of the assessed years are when there is at least some volcanic aerosol in the stratosphere. Saying "for up to several years after the eruptions" or similarly rewording would better convey the time frame this study focuses on.

We agree and now L16 reads:

"for up to several years following such events"

Lines 17-18: Comparing long-lasting effects from tropical eruptions to dynamical effects in high-latitude eruptions gives the impression that atmospheric dynamics is causing eruption effects to outlast the volcanic aerosol presence. Instead it is the slow ocean timescale for sea surface temperatures to re-equilibrate. I recommend rewording to avoid this confusion.

This is well spotted, and now reads (L16-18):

"Whereas the mechanisms responsible for the prolonged response to volcanic surface cooling have been extensively investigated for tropical eruptions, less is known about the dynamical response to high-latitude eruptions."

line 17: "long-lasting response" here is confusing, as this is easy to conflate with the "long-lasting" duration of the eruptions themselves mentioned in the title, which is a quite different feature.

We agree, where this has been addressed in the above comment.

Line 28-9: I find confusing the line "The identification of a deterministic response such as increased SSWs following HL volcanic eruptions calls for increased attention". I think it would be better to say what the ensemble here "suggests" impacts on SSWs and clarify that more simulations than used here are needed to rigorously establish this.

We are certain that our edit to the Abstract addresses this comment, since this sentence has been removed.

Lines 31-2: I find this line problematic: "sensitivity of such events to eruption magnitude needs to be evaluated in terms of a possible source of increased seasonal predictability of

NH regional climates". The results of this study show too much noise to expect noticeable benefit, contradicting this "need". Also, the part "sensitivity [...] to eruption magnitudes" is odd because a once-per-century or rarer event would be a very minor "source of [...] seasonal predictability", while more common eruptions will have little signal. I would omit this line or make a different sentence stating that/how the results further the understanding of regional climates that follow such an eruption.

Similar to the above, this sentence has been removed.

Lines 163-168: I find the description of Lake and Eldgja as "much larger" than the 14 Tg case examined here misleading, as global climate responses are primarily caused by the stratospheric aerosols, while the much larger tropospheric aerosol amounts not simulated here will have far less impact per mass, as they fall out of the atmosphere much more quickly.

We agree, "much larger" has now been removed from the sentence where it serves as a natural comparison to our idealized experiment

Lines 362: I cannot believe this tiny p-value, as the very varied number of SSWs in the control case show these results lack high confidence. There is not enough data to seriously show results from this test. I would omit this and let the plotted results speak for themselves.

That is true, the p-value has been removed from the sentence.

Lines 366: I don't believe that atmospheric dynamical response from one winter is "an important precursor to the significant increase in SSWs" a year later. The atmosphere does not have this kind of memory. If there is a connection between the two winters, it would be because sea surface temperatures maintain a similar post-eruption pattern, which in turn could cause similar atmospheric circulation patterns.

We agree and thank the reviewer for identifying this misstatement. This now reads (L458-461):

"This upward wave flux and weaker winds continue into the third winter, where winter 2 likely acts as a precursor, allowing for SSWs to develop more frequently as detected in the T50 warming that is now confined over the polar cap (Fig. 4c and Fig. 5c, respectively)."

Lines 386-7: Please word this important sentence more accurately and preferably walk the reader through it. The results here "suggest" an effect on SSWs, while a considerably larger ensemble would be needed to properly establish this.

We agree, where we feel that our additions related to RW1 comments like the SSW analysis for atm-only (that is also related to an earlier and much relevant comment from RW2) and the analysis on the potential impact on ensemble size on the signal to noise clarifies this part and adds more confidence (Fig. S7 and Fig. 7 respectively). This is described in L411-438:

"To better understand the cpl SSWs response, we also did an SSW analysis on *atm-only* (Fig.

S7) where 50-75% less SSWs were detected in the perturbed simulation compared to the

unperturbed one. Such a response should not be unexpected during the forced polar vortex strengthening as detected in *atm-only* (see Fig. 5). Furthermore, only single SSWs per winter were detected in all 20 ensemble members of the perturbed simulation while two (one) ensemble member(s) detected double SSWs per winter 1 (winter 2) in the unperturbed simulation. Although these results do show strong evidence of an increase in the number of SSWs in the *cpl* simulation, internal variability is large and the frequency of SSWs fluctuates substantially between the three winters in the unperturbed simulation.

Since this is indicative of a low signal-to-noise ratio and uncertainties in the response of polar vortex variability, we tested the potential impact of ensemble size on the signal-to-noise ratio for two key diagnostics of our mechanism, namely U10 and SSW. We express the signal-to-noise ratio as the uncertainty related to the expected (i.e., ensemble mean) response, calculated as standard error of the mean of post-eruption paired anomalies for the first three post-eruption winters. The standard errors converge toward the value obtained for the full 20-member ensemble for all winters and both variables (Figure 7), with a common tendency of growing uncertainty in the expected response with the ensemble size. The mean curves consistently level off for ensemble sizes larger than 10, suggesting that the full ensemble estimate is representative of uncertainty in the expected response for larger ensemble sizes. Otherwise, winter 3 produces larger uncertainty than winters 1 and 2, suggesting a less constrained forced response. This is especially evident for estimates of standard error of SSW anomalies and ensemble sizes larger than 15, where winters 1 and 2 closely superpose on each other while winter 3 does not overlap with winters 1 and 2 within the 5-95th percentile range. Overall, the flattening of the ensemble-mean expectations on the standard error, and the large values diagnosed in winter 3, suggest that the winter 3 response features an intrinsically lower signal-to-noise ratio. In fact, since the value of full-ensemble mean SSW paired anomaly in winter 3 (+1.05 events) is similar to the associated expected standard error (+1.02 events) we conclude that even a much larger ensemble would not provide more certainty in the signal detected. "

Discussions and Conclusions have also been changed accordingly.

Fig. 6: There are now two Fig. 6's. Please shift the number of the second Fig. 6 and all later plots, and mentions of these in the text, to correct this.
We have corrected this mistake.

Fig. 6 (second): The font size in all the sub-plots is far too small.
This has been updated and should be more clear now.

Line 660: "A significant increase" in SSWs should just be "an increase", as there is not enough data to reliably say this is statistically significant. We would need to know what the distribution of SSW counts is across many unperturbed years, and the 3 very different counts in each control year in Fig. 6d show that the current evaluation does not have enough.
Correct, we have now changed accordingly.

Reviewer #1 - Report #2

We thank the reviewer for these important comments. We are certain that our response has resulted in valuable improvements regarding the reliability of our results.
In addition to the reviewer comments and changes associated with those comments, we made three changes to the manuscript that was not part of the report comments.
Firstly, we added an explanation behind the LW/SW difference between cpl and atm-only. That is mentioned in subchapter 3.1. (L368-271):
"We also detect a slight difference in the LW and SW fluxes that arises from differences in high cloud cover between *atm-only* and *cpl*, where *cpl* shows a decrease in high cloud cover in the northern high-latitudes, compared to *atm-only* (not shown)."
and, in the Discussions (L676-681):
"Interestingly, a slight difference between *atm-only* and *cpl* is detected in both the LW and SW flux that is caused by a strong significant decrease in high cloud cover in the *cpl* simulation (not shown). This cloud cover decrease, especially at mid- to high latitudes, agrees with the increased LW fluxes at higher latitudes in addition to the decrease in SW flux and the associated surface cooling. This raises a question on the role of forced surface processes in these high cloud changes but since it is out of the current scope we leave it open for further studies."

Secondly, we also changed the scale in the Tgrad figure (Fig. 10) to improve clarity.

Finally, we added panels showing the tropospheric zonal wind anomalies at 200 hPa (U200) for cpl and atm-only to figures 8-9 in addition to a figure showing changes in waveguides (Figure 10) (shown as a probability of favorable propagation for Rossby waves). We did this since the nature of the strong tropospheric response detected and the equatorial movement of the subtropical jet should result in changes in wave activity. This we confirm, where we show that upward wave propagation is more favorable in winters 1-2 in addition to a waveguide decrease in the subtropical troposphere that also favors large-scale waves to be redirected towards the pole.
We added the following in the Tropospheric section (L527-555):
"Since upward wave activity depends on wave mean flow interactions, several factors are at play to explain the strong response in *cpl* vs *atm-only*. First, the change in zonal flow is substantially different between the two pairs of experiments, as shown by the U200 anomalies (Fig. 8g-i and Fig. 9e-f). In the first two winters we observe an intense deepening of the Aleutian low in *cpl* (Fig. 8a-b) associated with a large equatorward shift of the subtropical jet over the North Pacific (Fig. 8g-h, also seen in the zonal mean averages of Fig. 4). The change in zonal flow is not as large in *atm-only*, where there is a general decrease of U200 on the poleward side of the subtropical jets, rather than a marked equatorward shift as in *cpl*. This further emphasizes that amplified surface coupling when the ocean is coupled

to the atmosphere has a dramatic impact on the amplitude of the tropospheric response. Because the zonal flow acts as a waveguide for large-scale planetary waves, we expect changes in upward wave propagation in the stratosphere. To measure how waveguides change, Fig. 10 shows the probability of favorable propagation conditions for large-scale stationary waves, averaged for zonal wave numbers k=1,2,3 and meridional wave numbers l=1,2,3, in function of pressure and latitude (see section 2 for more detail). Areas of high probability show where large-scale waves preferentially propagate, while low probability regions indicate where wave propagation is prohibited. Generally, the mid-latitude troposphere is more favorable for wave propagation than the high-latitudes and the stratosphere, consistent with the tendency for stationary waves to propagate upwards and to be deflected towards the equator, in climatology. After injection of the volcanic forcing, both *cpl* and *atm-only* exhibit an increase in the probability for wave propagation between 40 and 60°N in the lower stratosphere during winter 1 and 2, but the responses in the troposphere are markedly different. In *atm-only*, wave propagation is inhibited in the free troposphere north of 60°N, for both winters 1 and 2 (Fig. 10d-e), which is consistent with the EP-flux anomalies of Fig. 5. This response is absent from *cpl* during winter 1 (Fig. 10a), and opposite during winter 2 when an increase of favorable conditions for wave propagation is observed (Fig. 10b). We also see that the waveguide has greatly reduced in the subtropical troposphere in *cpl* winters 1-2 that favor large-scale waves to be redirected towards the pole. This increase in favorable conditions for wave propagation in the troposphere between 60 and 80°N persists during winter 3 in *cpl* (Fig. 10c), which is a partial explanation for enhanced upward wave propagation in the stratosphere described in Fig. 4f."

We also added to the Abstract, Methods, Discussions and Conclusions accordingly.
Since this is much more significant to our study, we have shortened the eddy-feedback part and put the figure in SI.

Then the reviewer's comments (in black) and the associated response (in red) are listed here below:

Some major questions and concerns raised in previous comments still exist in the revised manuscript. For example, the emphasize on the "long-lasting" emission and limited ensemble members to investigate polar vortex and SSWs change, and 2 to 3 SSWs in one single winter simulated by the model.

We have now removed "long-lasting" from the title and replaced it with "large" that should decrease the emphasis on our experiment being "long-lasting". We also feel that the paragraph here below (L168-176) clarifies this:

"We thus obtain aerosol optical properties for an idealized, long-lasting high-latitude NH eruption. In this experiment we assume stratospheric injection only, although similar eruptions in the natural world would likely inject part of the total aerosol mass within the troposphere during the eruption. Past NH eruptions like Eldgjá and Laki had an atmospheric $SO_2$ loading of 219Tg and 122Tg respectively that was carried aloft with the eruptive column up into the upper troposphere with portions of the aerosols reaching the lower stratosphere during the eruptions (Thordarson et al., 2001). Hence our experiment can also be considered as a 6-month stratospheric aerosol injection that is analogous to similar although smaller eruptions (as compared to Laki) without the tropospheric aerosols."

Also, in the main text where we discuss our results, we now refer to an enhancement of stratospheric aerosols that should also contribute to this clarification.

The following paragraph on our recently added SSW analysis for atm-only has been added to the Results section 3.2.3 in addition to the addition of the associated Figure in the Supplementaries (L411-419):

"To better understand the cpl SSWs response, we also did an SSW analysis on *atm-only* (Fig. S7) where 50-75% less SSWs were detected in the perturbed simulation compared to the unperturbed one. Such a response should not be unexpected during the forced polar vortex strengthening as detected in *atm-only* (see Fig. 5). Furthermore, only single SSWs per winter were detected in all 20 ensemble members of the perturbed simulation while two (one) ensemble member(s) detected double SSWs per winter 1 (winter 2) in the unperturbed simulation. Although these results do show strong evidence of an increase in the number of SSWs in the *cpl* simulation, internal variability is large and the frequency of SSWs fluctuates substantially between the three winters in the unperturbed simulation. "

We also added the following to the Discussions (L642-654):

"We also see that the large decrease in SSWs in the perturbed simulation of *atm-only* (when compared to unperturbed) is consistent with the detected polar vortex strengthening. This further supports the significance of the signal we detect in *cpl* winter 3 compared to the background noise. In addition, all winters examined, in both *cpl* and *atm-only*, showed that there is up to 15% chance of getting more than 1 SSWs per winter in all ensemble members. This is not far from Ineson et al. (2022) who identified a double event once every 9 years in a

66-year ERA5 record. The exception is *cpl* winter 3 that is also the only winter that has 3 SSWs, with the average SSW occurrence also being the only winter above 1 (1.17) while all other winters span between 0.15-0.85 per winter. A similar NH high-latitude eruption has not taken place during the observational period, so we have no comparison. Also, to the best of our knowledge, a similar high-latitude sulfur injection study has not been performed before. Therefore, it is difficult to say at this stage if such a response is realistic or not, but in general more than two SSWs per winter can be considered exceptional yet plausible, as is also the case for our idealized eruption."

Additionally, we added analysis to test the impact of the ensemble size on the signal-to-noise ratio. This is depicted in Figure 7 and the following text was added (L420-438):
"Since this is indicative of a low signal-to-noise ratio and uncertainties in the response of polar vortex variability, we tested the potential impact of ensemble size on the signal-to-noise ratio for two key diagnostics of our mechanism, namely U10 and SSW. We express the signal-to-noise ratio as the uncertainty related to the expected (i.e., ensemble mean) response, calculated as standard error of the mean of post-eruption paired anomalies for the first three post-eruption winters. The standard errors converge toward the value obtained for the full 20-member ensemble for all winters and both variables (Figure 7), with a common tendency of growing uncertainty in the expected response with the ensemble size. The mean curves consistently level off for ensemble sizes larger than 10, suggesting that the full ensemble estimate is representative of uncertainty in the expected response for larger ensemble sizes. Otherwise, winter 3 produces larger uncertainty than winters 1 and 2, suggesting a less constrained forced response. This is especially evident for estimates of standard error of SSW anomalies and ensemble sizes larger than 15, where winters 1 and 2 closely superpose on each other while winter 3 does not overlap with winters 1 and 2 within the 5-95th percentile range. Overall, the flattening of the ensemble-mean expectations on the standard error, and the large values diagnosed in winter 3, suggest that the winter 3 response features an intrinsically lower signal-to-noise ratio. In fact, since the value of full-ensemble mean SSW paired anomaly in winter 3 (+1.05 events) is similar to the

associated expected standard error (+1.02 events) we conclude that even a much larger ensemble would not provide more certainty in the signal detected."

According to the above, we also added to Discussions (L640-642):
"Furthermore, the SSW analysis for *atm-only* and the ensemble size test (Fig. 7) both show strong evidence of a robust signal for winter 3 despite the noisy polar vortex and the limited ensemble size."

Besides, some long sentences can be separated into shorter sentences to increase the clarity.
We thank the reviewer for this important comment. We have now revised the manuscript with this in mind where we have shortened most of the sentences the reviewer refers to.

Except for the unsolved major concerns, below are some minor comments:
Title: "long-lasting". As commented by reviewers, this can be misleading and considering that there were eruptions that is much longer than 6 months as shown in this paper: Gabriel, I., Plunkett, G., Abbott, P.M. et al. Decadal-to-centennial increases of volcanic aerosols from Iceland challenge the concept of a Medieval Quiet Period. Commun Earth Environ 5, 194 (2024). https://doi.org/10.1038/s43247-024-01350-6
We thank the reviewer for this comparison. However we do feel that this paper is not directly comparable to ours since it is about the Icelandic Active Volcanic period where several volcanic systems contributed to the aerosol loading - most of them being tropospheric like the Hrafnkatla eruption that had one stratospheric event. With that being said, we agree that only using "long-lasting" can be confusing, where our experiment simulates long-lasting stratospheric eruption/aerosol injection. We have thus made changes to the text accordingly in addition to removing "long-lasting" from the title.

Line 159-160: October 1 should be November 1 if starts from May 1 and lasts for 6 months.
Correct, this has been fixed.

Line 163-164: atmospheric loading? Better specify SO2 or SO4, that's very different. Are the loading numbers correct based on the reference? In a new paper Hutchison et al., 2024, it writes "The Eldgjá eruption... estimated to have released 220 Tg of SO2 into the atmosphere (with ~185 Tg of this reaching upper tropospheric and lower stratospheric altitudes, Thordarson et al., 2001)." 120Tg is a huge difference as written in this manuscript. Hutchison, W., Gabriel, I., Plunkett, G., Burke, A., Sugden, P., Innes, H., et al. (2024). High-resolution ice-core analyses identify the Eldgjá eruption and a cluster of Icelandic and trans-continental tephras between 936 and 943 CE. Journal of Geophysical Research: Atmospheres, 129, e2023JD040142. https://doi.org/10.1029/ 2023JD040142
We thank the reviewer for correcting this misstatement, the eruption order had been mixed. It now writes (L163-164):
"Past NH eruptions like Eldgjá and Laki had an atmospheric $SO_2$ loading of 219Tg and 122Tg respectively,"

Figure 1: still not clear. Previous reviewers' comments gave suggestions which should be helpful but was not adopted.

We agree and now this figure has been updated where each forcing has its own axis. We also took the opportunity to update the figure since the forcing curves ended on 1. Dec but not 31. Dec as they should have. Hopefully this adds more clarity.

L195-197 and figure 1-3: Figure 1 looks correct with three post-eruption winters. But when looking at figure 2 and figure 3, there is only two winters after the eruption? The explanation to subsequent figures should be connected to figure 2 and 3, then they need to be double checked and clarified.

This is true, Figure 1 shows the 36 month long volcanic forcing profile used in our simulations. However, as can be seen in Fig. 1 the aerosols have decreased substantially in the 4th year and since the memory in the atmospheric component is very small, the atmosphere-only simulation only ran for 3 years - thus explaining why the third winter is absent in atm-only. This is mention here in lines 190-192:

"The atmosphere-only experiments were run over three full years, which provides two full winters after the onset of the eruption. We found that there was no need to extend the simulations further given the duration of the forcing and short memory of the atmosphere."

In the coupled simulation we use the same forcing and allow it to run for a longer time since there the memory comes into play.

L223-224 what temperature gradient is this? Needs to be clarified. Different from stratospheric temperature "gradient" in L40?

This would be Kelvin, this has been added in the text.

L234: decrease in the LW? It's confusing. Better change negative value to positive value of different direction to SW and adjust the description in the manuscript.

We agree, this is confusing. We have changed the LW colorbar so it now shows positive values and the text has changed accordingly.

L244-245: radiative forcing confined to NH extratropical summers?

We thank the reviewer for spotting this error, this now reads (L262-263):

"Overall, the radiative forcing is largely bounded by the NH extratropics..."

Figure 6: It would be nice to show the evolution of the SSWs as suggested by reviewer 2, as it's still not clear how you counted the numbers and doubtful about 2 to 3 SSWs in one winter.

We feel this has been addressed in the first comment in addition to showing significant U10 winds in Fig. 3c, where the zonal wind decrease becomes clear for *cpl*. Where in addition to showing SSWs for atm-only, we also show that although 2 SSWs per winter are not common - they do occur in reanalysis datasets spanning 66 years (in total 9 events) in addition to observations (see references in main text). It is the 3 SSWs per winter that is unprecedented - and since it only occurs in winter 3 it is more likely that the forced response is responsible - especially when comparing the SSW statistics between cpl and

atm-only. If we remove the two cpl ensemble members that show 3 SSWs to treat as outliers, the SSW increase is still significant (not shown).

Figure 7: line labels not clear.
True indeed, this should be better now.

L1089-1096 reference format needs to be double checked, these two references, one has the publish year after the authors while the other one is at the end. Also exist in several other references.
This has now been checked and edited in accordance with the journal reference guidelines.

L1098-1100 "eaat6025. doi:10.1126/sciadv.aat6025. PMID: 30050990; PMCID: PMC6059732." ?
Thank you for noticing, this reference has been updated.

L1105-1106 The citation you used in the discussion part, better check the published paper:

Zhuo, Z., Fuglestvedt, H. F., Toohey, M., and Krüger, K.: Initial atmospheric conditions control transport of volcanic volatiles, forcing and impacts, Atmos. Chem. Phys., 24, 6233–6249, https://doi.org/10.5194/acp-24-6233-2024, 2024.
Thanks, done.

---

## Author Response (AR3)

Reviewer #1 (report #2)

The manuscript is greatly improved after adding analysis on the comparison between cpl and atm-only, the impact of ensemble size and the tropospheric response. Although it's hard to verify the reliability of three SSWs in one winter, the results do show a clear difference of winter 3 compared to winter 1 and 2. And the mechanisms explored in this study should add extra value to the research topic. The language can still be improved with a proof reading. The study is worth publication after addressing a few remaining issues as commented below.

We thank the reviewer for their detailed and much relevant revision during this process that has contributed to the accuracy of this work. Here below are our response to the latest reviewer's comments in red. Line numbers refer to the track changes version of this manuscript.

L29: an potential increase? Since it's hard to verify the three SSWs in one winter.
Indeed, we have altered the sentence accordingly.

L57: "play a deciding role" really?
Since we further explain this statement in that paragraph we are confident of this wording, although we are aware that the polar vortex is not the only player as we also discuss in the text.

L187-189: Confusing sentence, better to rephrase. Is it the SSWs play a role in the atmospheric circulation response?
We agree that is can cause confusion and have changed lines 130-132 according to the following:
"This is the focal point of this study where we investigate for the first time the role of wave-mean flow interactions and SSWs in the climate response to a HL volcanic eruption."

Figure 2: It's a bit confusing to show darker color for larger anomaly for SW, but darker color for smaller anomaly for LW. Better to reverse one to keep it consistent.
Ok, we have changed back to the original version.

Figure 1 and 2: "aerosol mass" is confusing, as kg/kg is for mass mixing ratio, not mass. Please double check what's exactly used in the model, what is shown in figure 1 and 2 and use the correct text. Please also check texts written in the manuscript. Figure 1 why starts from month 5 instead of from 0?
We thank the reviewer for this well spotted comment. Indeed it is important to define between these two parameters in more details than we did. Kg/kg represents dry volcanic aerosol mass mixing ratio while kg/m2 represents average aerosol column mass. We have edited the text accordingly (L180, 183, 202) and changed figures appropriately.
Regarding Figure 1 starting from month 5, we mainly want to draw the attention towards the different profiles used for scaling the final WACCM input. Also, unlike Figure 2, Figure 1 does not show a model output. Also, since the values in months 1-4 are zero it would make the figure less clear starting the x-axis in month 1 (january).

Figure 2 and 3: If cpl runs for 4 years and atm-only runs only for 3 years, then it's clear and consistent to show four years for both and leave the 4th year blank for the atm-only. Especially, the aerosol distribution and U10 variations in the 3rd post-eruption winter in Fig. 3 should also be shown.

We do agree and have edited Fig. 3 accordingly. In addition, we took the advantge to increase the number of contours to show more details. In stead of leaving the 4th year black, we underline its different time axis in the figure caption.

We also added the following sentence (L531-533):

"The SSW development is also evident in both U10 and U50 and T50 timeseries (Fig. 3c and Figure S3a-b respectively), where peak T50 warming occurs late in winter 3."

We have also edited the following sentence to explain further the volcanic forcing used for winter 3 (L231-232):

"...,where the January and February forcing of winter 3 are defined to be a continuation of the December value of year 2"

We did not want to complicate part 2.3 but of course it is important to mention the forcing in winter 3 in more details although we do not see the need for an extended Figure 2.

L259-261: For a clear comparison, what is the total loading of SO2 in Tg in your simulations?

We have added the following to the sentence in L294-295 for clarity:

"and the total aerosol mass of 14.04 Tg being largely confined north of 45° N (Fig. 2a-b)."

L262: the radiative forcing is SW, LW or you mean the idealized volcanic forcing?

Indeed, this now reads idealized (L295).

L427-438: Better rephrase this. Figure 7 supports that winter 3 is a less constrained response compared to winter 1 and 2, but does not necessarily mean a much larger ensemble would not provide more certainty. The results only show that the standard error will decrease (increase) when ensemble of more (less) members compared to the ensemble of 20. But How's it when create ensemble of members more than 20. And analysis in Figure 7 is based on these same ensemble members, and but with different new members results may be different?

We agree where this comment is also related to the comment from reviewer #2 and we refer to our response regarding a more detailed response. We have moved this Figure into supplementary and shorten the text that now reads (L462-467):

"We explored the impact of the ensemble size for the ensemble spread of two key diagnostics of our mechanism, namely U10 and SSW, calculated as the standard deviation of post-eruption paired anomalies for the first three post-eruption winters (Supplementary Fig. S8). Winter 3 produces larger spread than winters 1 and 2, indicative of a least constrained forced response, which is especially evident for ensemble sizes larger than 15. Therefore, only much larger ensembles may provide signals not encompassing the value of zero within uncertainty."

Figure 8: i) DJF3 not DJF2. The figure caption should be rephrased for better clarity.

Thanks, done.

L557-558 and Figure S5: It's confusing. From Figure 1 and 2, it looks winter 1 match Jan01, winter 2 match Jan02, right? Then winter 3 should match Jan03? But Figure S5 only goes to Jan02, then one cannot see which is evident in the SST.

Indeed the reviewer is correct, it is better to show winter 3, we have now updated supplementary Figure S5.

L626: "that we further confirm to occurr" better rephrase.

Thanks, this now reads (L763-764):

"...,agreeing with our results in winter 3 where an increase in the occurrence of SSWs is detected."

L648-654: Can you add a (supplement) figure to show the zonal wind and temperature timeseries of members with three SSWs in winter 3. With only the count in figure 6 and S3, it's hard to know how the SSWs look like. It's hard to know if it's realistic or not, then better to show the evolution in the paper, then if any future study shows three SSWs in one winter, this opens a chance for future comparisons.

This is related to the comment above, where we do agree and have updated both Figure 3 and Figure S3 accordingly, so now the U10, U50 and T50 time series show this evolution. We note however that the temperature pattern at 50hPa (see Figure S3) is spatially different when compared to the zonal wind, where the warming is mostly confined in part of the NH. In short, the asymmetric nature of the warming pattern causes a dampening of the response when averaged over the entire NH. Therefore the T50 timeseries are based on 70-90°N and 0-200°E compared to 70-80°N and 0-360° used for U50.

References: doi links should be added for all the references?

Ok, done (no doi found for two papers).

Reviewer #2, Report #1

I felt this study was nearly ready for publishing, although now the authors have added a problematic figure that seems to convey confidence in a large SSW effect despite too few simulations to address the extremely noisy occurrence of SSWs. If this analysis cannot be thoroughly defended, I request that the new figure and added statements professing confidence in the SSW results please be removed, in which case my review would return to being minor concerns.

We thank the reviewer for their detailed and much relevant revision during this process that has contributed to the accuracy of this work.Here below are our response to the latest reviewer's comments in red. Line numbers refer to the track changes version of this manuscript.

- The new Fig. 7 appears to be giving a misleading argument that 20 ensemble members is sufficient to validate a strong SSW signal when this is not being demonstrated. The possible iterations of <20 ensemble members converge as ensemble size increases toward the full 20-member ensemble, but not because the uncertainty is small. By n=10, nearly any two random 10-member combinations of the full 20-member ensemble will share at least a few common members, and hence convergence toward the mean of the full ensemble is expected. We really need more ensemble members to be confident of an SSW impact in Winter 3. From the noisy data in Fig. 6, which shows very difference mean SSW numbers among unforced seasons that are statistically equal, I expect a different set of 20 simulated eruption realizations could result in an entirely different conclusions, and this new analysis does nothing to change that impression. The information that matters is already presented in Fig. 6, so I don't see this Fig. 7 being anything but a distraction from the more interesting features of this study.

We thank the reviewer for noticing this. There were indeed mistakes in the caption of the figure, which illustrates changes in uncertainty of the ensemble response (standard deviation), not changes in the ensemble mean, which inevitably led to confusion regarding our interpretation. In fact, the key points we made in the main text are in line with the reviewer's interpretation: we observe an increase in uncertainty with ensemble size and especially for winter 3, which implies that, as we wrote, "winter 3 produces larger uncertainty than winters 1 and 2, suggesting a least constrained forced response" and "winter 3 features an intrinsically lower signal-to-noise ratio". The mistake in the caption was corrected, and we are confident that our statements stand. We also agree that Fig. 7 adds little information with respect to what is shown in Fig. 6, so we decided to move it to the supplement. We have also reduced the description of the results in the main text, focusing on the main implications of the analysis that now reads (L462-467): "We explored the impact of the ensemble size for the ensemble spread of two key diagnostics of our mechanism, namely U10 and SSW, calculated as the standard deviation of post-eruption paired anomalies for the first three post-eruption winters (Supplementary Fig. S8). Winter 3 produces larger spread than winters 1 and 2, indicative of a least constrained forced response, which is especially evident for ensemble sizes larger than 15. Therefore, only much larger ensembles may provide signals not encompassing the value of zero within uncertainty."

We also rephrased the sentence in line 466-467 as follows: " Therefore, only much larger ensembles may provide signals not encompassing the value of zero within uncertainty "

- Also on Fig. 7, it seems odd that the mean among all permutations should be anything but constant as a function of sample size. Every set of permutations would be expected to have each ensemble member represented an equal number of times, and hence equal the mean of the full ensemble. So the statement that the "mean curves consistently level off for ensemble sizes" is entirely mysterious to me, and seems to make no proof of the 20-member ensemble being representative of a potentially larger ensemble.

This comment follows our mistake in the captioning and labelling of the figure (corrected in the revised version). We confirm that, of course, the analysis for the ensemble mean leads to a constant value across all ensemble sizes. Figure 1 here below (ensemble mean analysis) provides proof of this.

[Figure]

Figure 1: The same as Figure S8 but for the ensemble mean value

- The text in Lines 416-438 that accompanies Fig. 7 makes strong statements that I find disagreeable, i.e. "these results do show strong evidence of an increase in the number of SSWs" and "the full ensemble estimate is representative of uncertainty in the expected response for larger ensemble sizes". I do not see sufficient evidence for either conclusion. I also find the wording of this section generally confusing.

line 416-417 (now 459): we have changed " show strong evidence of" with "suggest"
line 418-438: this was deleted.

- The phrase "paired anomalies" is used throughout the text but is not defined anywhere.

This we already defined in lines 246-248:

"Model output is analyzed by computing paired anomalies, defined as deviations of each volcanic simulation from the corresponding control simulation (Zanchettin et al., 2022) (volcanic minus control)."

- Of the eruption effects, Lines 28-29 of the Abstract states that "This causes a weakening of the polar vortex and an increase in the occurrence of sudden stratospheric warming events, although with a small signal-to-noise ratio". This still sounds like there is a cause-and-effect link that hasn't been convincingly demonstrated. I'd prefer if this line instead says that simulations with eruptions are found to have an unusually high frequency of SSWs and this warrants further exploration. While I'm only addressing in this review that the signal-to-noise issue has not been resolved, there are other the issues, i.e. that this is a single-model study and SSWs are notoriously difficult to reliably simulate in climate models, so I stress not to place too much confidence in the SSW results.

We agree and now this sentence reads (L29-30):
"We detect unusually high frequency of SSWs in the idealized forcing simulation using interactive ocean that calls for further exploration."

- Line 459: "winter 2 likely acts as a precursor" to "potentially acts" or "seems to act", please.
Indeed, "potentially" has been added to (now) L479.

- Some aspects of the title that the other reviewer and I both disliked have now been removed, and I thank the authors for addressing these concerns. Looking at the now quite short title, I feel it could be improved by specifying that this is a global modeling study, i.e. adding to the title's end something like "in a global climate model" or "in CESM1-WACCM4"
Yes that would be relevant to make such additions, now the title reads:
"Stratospheric circulation response to large Northern high-latitude volcanic eruptions in a global climate model"

- I do not understand Lines 99-100: "In the definition of a framework to study the climatic effects of a high-latitude enhancement of the stratospheric sulfate aerosol layer, Icelandic volcanism provides for an ideal test bed" could just be "Icelandic volcanism is an important case study for high-latitude eruption impacts on climate".
We see the reviewer's point and have removed this part. Now this reads (L116-117):
"Icelandic volcanism has played a role in shaping past NH climate variability and will continue doing so."

- Some wording suggestions to spruce up the abstract: First, the opening sentence is quite wordy, which makes it hard to read. I would simplify "The temporary enhancement of the stratospheric aerosol layer after major explosive volcanic eruptions" to just "Stratospheric aerosols from major explosive volcanic eruptions" or similar. Second, Line 21's instance of "with an interactive ocean, or with prescribed [...]" to "with an interactive ocean and with prescribed [...]".
We thank the reviewer for their detailed comments, we the first sentence now reads:
"Stratospheric aerosols after major explosive volcanic eruptions can trigger climate anomalies for up to several years following such events."

---

## Author Response (AR4)

Reviewer #2, Report #1

We are very grateful for the reviewer's much valid requests. We have now addressed them here below in red.
* * *
I thank the authors for addressing my comments. I feel this is back on track to publication and have only a few short requests.

Lines 605-7: My biggest remaining concern is these lines: " Furthermore, the SSW analysis for atm-only and the ensemble size test (Fig. S7 and Fig. S8 respectively) both show strong evidence of a robust signal for winter 3 despite the noisy polar vortex and the limited ensemble size." I disagree with "strong evidence" for reasons given by both reviewers previously, and request (as with other lines that have already been revised) that this be edited to "suggests" or similar.
Indeed we agree on the careful wording here in agreement with the revised part based on exactly that. Now lines 611-613 reads:
"Furthermore, the SSW analysis for *atm-only* and the ensemble size test (Fig. S7 and Fig. S8 respectively) both suggest the presence of a robust signal for winter 3 despite the noisy polar vortex and the limited ensemble size."

Line 30: "in the idealized forcing simulation with interactive ocean" is hard to understand, and really there's many simulations (i.e. an ensemble). I find "idealized forcing" here confusing and unneeded, as this isn't significantly more idealized than is usual in a GCM. "in simulations with interactive ocean temperatures" would maybe suffice.
We agree, this sentence now reads:
"We detect unusually high frequency of Sudden Stratospheric Warmings in the simulations with interactive ocean temperatures that calls for further exploration."

Line 417: "a least constrained forced response" to "a less constrained forced response"?
Correct, Line 418 now reads "less constrained".

Lines 418-9: "Therefore, only much larger ensembles may provide signals not encompassing the value of zero within uncertainty." This isn't clearly saying the main point, so could be rewritten to correct this. E.g. something along the lines of "This analysis suggests that larger ensembles would be needed to demonstrate high confidence in the SSW response (i.e. signals not encompassing the value of zero within uncertainty)".
Indeed we agree that this line is somewhat confusing. Now lines 419-421 reads:
"Accordingly, this analysis suggests that much larger ensembles are needed to confidently demonstrate the significance of the SSW response (i.e. to provide signals not encompassing the value of zero within uncertainty)."